# Dysfunctional S1P/S1PR1 signaling in the dentate gyrus drives vulnerability of chronic pain-related memory impairment

Mengqiao Cui[1,2,3†], Xiaoyuan Pan[1,2,3†], Zhijie Fan[1,2,3†], Shulin Wu[1,2,3], Ran Ji[1,2,3], Xianlei Wang[1,2,3], Xiangxi Kong[1,2,3], Zhou Wu[1,2,3], Lingzhen Song[1,2,3], Weiyi Song[1,2,3,4], Jun-Xia Yang[1,2,3], Hongjie Zhang[5], Hongxing Zhang[1,2,3], Hai-Lei Ding[1,2,3]*, Jun-Li Cao[1,2,3,6]*

[1]Jiangsu Province Key Laboratory of Anesthesiology, Xuzhou Medical University, Xuzhou, China; [2]NMPA Key Laboratory for Research and Evaluation of Narcotic and Psychotropic Drugs, Xuzhou Medical University, Xuzhou, China; [3]Jiangsu Province Key Laboratory of Anesthesia and Analgesia Application Technology, Xuzhou Medical University, Xuzhou, China; [4]School of Public Health, Xuzhou Medical University, Xuzhou, China; [5]Faculty of Health Sciences, University of Macau, Taipa, China; [6]Department of Anesthesiology, the Affiliated Hospital of Xuzhou Medical University, Xuzhou, China

*For correspondence:
haileimdar@yahoo.com (H-LD);
caojl0310@aliyun.com (J-LC)

†These authors contributed equally to this work

Competing interest: The authors declare that no competing interests exist.

## eLife Assessment

This study investigates the molecular mechanisms underlying chronic pain-related memory impairment by focusing on S1P/S1PR1 signaling in the dentate gyrus (DG) of the hippocampus. Through behavioral tests (Y-maze and Morris water maze) and RNA-seq analysis, the researchers discovered that S1P/S1PR1 signaling is crucial for determining susceptibility to memory impairment, with decreased S1PR1 expression linked to structural plasticity changes and memory deficits. This work has **important** significance and a **convincing** level of evidence, thus offering new insights into the mechanisms underlying chronic pain-related memory impairment.

**Abstract** Memory impairment in chronic pain patients is substantial and common, and few therapeutic strategies are available. Chronic pain-related memory impairment has susceptible and unsusceptible features. Therefore, exploring the underlying mechanisms of its vulnerability is essential for developing effective treatments. Here, combining two spatial memory tests (Y-maze test and Morris water maze), we segregated chronic pain mice into memory impairment-susceptible and -unsusceptible subpopulations in a chronic neuropathic pain model induced by chronic constrictive injury of the sciatic nerve. RNA-Seq analysis and gain/loss-of-function study revealed that S1P/S1PR1 signaling is a determinant for vulnerability to chronic pain-related memory impairment. Knockdown of the S1PR1 in the dentate gyrus (DG) promoted a susceptible phenotype and led to structural plasticity changes of reduced excitatory synapse formation and abnormal spine morphology as observed in susceptible mice, while overexpression of the S1PR1 and pharmacological administration of S1PR1 agonist in the DG promoted an unsusceptible phenotype and prevented the occurrence of memory impairment, and rescued the morphological abnormality. Finally, the Gene Ontology (GO) enrichment analysis and biochemical evidence indicated that downregulation of S1PR1 in susceptible mice may impair DG structural plasticity via interaction with actin cytoskeleton rearrangement-related signaling pathways including Itga2 and its downstream Rac1/Cdc42 signaling and Arp2/3 cascade. These results reveal a novel mechanism and provide

a promising preventive and therapeutic molecular target for vulnerability to chronic pain-related memory impairment.

## Introduction

Clinical and preclinical studies have demonstrated that chronic pain impairs memory or accelerates memory decline (*Whitlock et al., 2017*; *Phelps et al., 2021*; *Zhao et al., 2023*). Memory impairment is also a potential contributing factor to the maintenance of chronic pain and poor treatment response (*Phelps et al., 2021*; *Wiech, 2016*; *Moriarty et al., 2011*). Although memory deficits in chronic pain patients are substantial and common, some clinical observations indicate high heterogeneity (*Whitlock et al., 2017*; *Innes and Sambamoorthi, 2020*), suggesting that chronic pain-related memory impairment exhibits susceptible and unsusceptible features. However, few preclinical studies model this clinical scenario and explore its underlying mechanisms.

Findings from both human and animal studies have indicated that cognitive dysfunction associated with chronic pain is linked to structural and functional deficits within the hippocampus (*Tajerian et al., 2018*; *Tajerian et al., 2014*; *Xia et al., 2020*; *Zhang et al., 2021*; *Mutso et al., 2012*). In particular, the dentate gyrus (DG), as part of the hippocampus, plays a crucial role in memory formation processing (*Zhang et al., 2008*; *Hainmueller and Bartos, 2020*). The DG has been postulated to perform a variety of mnemonic tasks, such as pattern separation (*Gilbert et al., 2001*), novelty detection (*Hunsaker et al., 2008*), and processing information related to spatial contexts (*Lee and Jung, 2017*). Nevertheless, the impact of pain-related cognitive syndromes on the dendritic morphologies of DG neurons, such as dendritic complexity, is inconsistent across different studies (*Tajerian et al., 2018*; *Tajerian et al., 2014*; *Tyrtyshnaia and Manzhulo, 2020*), and the molecular mechanisms remain minimally understood. The principal cell type of the DG is the dentate granule cells (DGCs), which are divided into immature and mature ones (*Amaral et al., 2007*). Immature newborn DGCs undergo neurogenesis and play key roles in learning and memory due to their high excitability and enhanced synaptic plasticity (*Kempermann et al., 1997*; *Ninkovic et al., 2007*). Our previous study found that mice with chronic pain-related memory impairment showed significantly reduced adult neurogenesis in the DG (*Xia et al., 2020*). Contrarily, mature DGCs are less excitable and exhibit reduced synaptic plasticity to an extent, but emerging evidence suggests that this cell population is equally recruited in memory formation (*Tronel et al., 2015*; *Liu et al., 2012*; *Ramirez et al., 2013*; *Redondo et al., 2014*; *Ryan et al., 2015*). It is intriguing to understand how mature DGCs would change in the state of chronic pain and by what molecular mechanism they facilitate chronic pain-related memory impairment.

Sphingosine 1-phosphate (S1P) is a bioactive sphingolipid metabolite, functioning as a key signaling molecule in a variety of cellular processes, such as cell division, adhesion, migration, and death (*Spiegel et al., 1996*; *Spiegel and Milstien, 2003*; *Cui et al., 2022*). S1P acts both through extracellular and intracellular modes (*Spiegel and Milstien, 2003*; *Cui et al., 2022*; *Maceyka et al., 2012*). In its extracellular mode, increasing evidence has suggested that the S1P system is a modulator of pain and memory processing pathways through S1P receptors 1–5 (S1PR1–5), particularly S1PR1. For instance, in models of traumatic nerve injury, astrocyte-mediated S1PR1 neuroinflammation contributes to central sensitization through increased S1P production and IL-1β release in the spinal cord, and beneficial effects of pain alleviation can be observed following IL-10-dependent S1PR1 antagonism (*Chen et al., 2019*). While there is evidence on S1P signaling regulating pain perception in the spinal cord, its role in higher pain centers remains largely unexplored. In regard to S1P function in memory processing, a recent study has hinted at the possibility of CNS S1PR1 agonism in the cellular repositioning of new DGCs and in regulating the integration of new neurons into pre-existing circuits, which may govern the process of memory formation (*Yang et al., 2020*). Additionally, activation of S1PR1 after traumatic brain injury in rats can significantly enhance neurogenesis and neurocognitive function (*Ye et al., 2016*). Importantly, increasing evidence suggests that the S1P receptor signaling pathway has profound effects on the regulation of synaptic strength, including modulating synaptic architecture and plasticity, and mediating excitatory synaptic transmission in the hippocampus (*Kanno et al., 2010*; *Riganti et al., 2016*; *Mitroi et al., 2016*). However, there is limited understanding regarding the involvement of S1P signaling in chronic pain-related memory impairment, as well as the interactive pathways that may underlie the effects.

In the present study, by employing an array of techniques including rodent-based behavioral tests, RNA-Seq, imaging, pharmacological and biochemical approaches, we elucidated a pivotal role of S1PR1 within the hippocampal DG in the context of chronic pain-associated memory impairment. Memory impairment-susceptible mice exhibited decreased S1PR1 expression in the hippocampal DG. Knockdown of S1PR1 in the DG facilitated the development of a vulnerable phenotype and led to abnormal structural plasticity in DGCs. Conversely, overexpression of S1PR1 or pharmacological administration of an S1PR1 agonist in the DG, promoted an unsusceptible phenotype, thereby averting the onset of memory impairment, and alleviated morphological abnormalities in DGCs. Subsequent mechanistic investigation demonstrated that, loss of S1PR1 in the DG results in actin dysregulation via interaction with integrin α2 (ITGA2) and its downstream Rac1/Cdc42 signaling and Arp2/3 cascade, leading to abnormal structural synaptic plasticity and ultimately causes memory impairment. Taken together, this study identified potential molecular mechanisms and promising therapeutic targets for preventing and treating vulnerability to memory impairment associated with chronic pain.

## Results

### Segregation of chronic pain mice into memory impairment-susceptible and -unsusceptible subpopulations

We utilized sciatic nerve chronic constriction injury (CCI) in C57BL/6J mice to model chronic pain state and tested the memory performance of mice with chronic neuropathic pain. CCI-induced long-lasting mechanical allodynia (*Figure 1—figure supplement 1A*) and thermal hyperalgesia (*Figure 1—figure supplement 1B*). No significant change in locomotor functions was observed in these injured mice (*Figure 1—figure supplement 1C, D*). Consistent with our recent study (*Xia et al., 2020*), only chronic (21–28 days, referred to as CCI-Chronic) and not acute (5–7 days, referred to as CCI-Acute) exposure to constrictive injury-induced neuropathic pain impaired spatial memory formation in both Y-maze test (*Figure 1A, B, D, E*) and Morris water maze (MWM) test (*Figure 1A, C, F, G*).

To investigate whether memory impairment has susceptible and unsusceptible features in CCI-induced chronic neuropathic pain mice, we analyzed results from Y-maze and MWM tests using the *k*-means clustering algorithm and segregated into two clusters in a large number of CCI mice (*k* = 2). After assigning each data point to its closest *k*-center, we drew a median between both the centroids as a cutoff value. One cluster, including mice displaying a ratio of time more than the cutoff value, was defined as the unsusceptible mouse cluster. The other cluster, including mice exhibiting a ratio of time less than the cutoff value, was defined as the susceptible mouse cluster. Using this algorithm, a ratio of 40% (percent time spent in the novel arm) and a ratio of 36% (percent time in the target quadrant) were set as cutoff values for Y-maze test (*Figure 1E*, left) and MWM test (*Figure 1G*, left), respectively. Accordingly, in the Y-maze test, 43% of CCI-Chronic mice were unsusceptible, and the remaining 57% were susceptible (*Figure 1E*, right). In the MWM test, 52% of CCI-Chronic mice were unsusceptible, and 48% were susceptible (*Figure 1G*, right).

We then used a four-quadrant chart to plot the data of CCI-Chronic mice memory performance for examining the consistency between the two behavioral assays. One variable, the percent time in the quadrant, was represented on the *x*-axis, and another variable, the percent time in the novel arm, was represented on the *y*-axis. The quadrants are determined by dividing the chart into four parts based on the cutoff values of Y-maze and MWM tests. As shown in the chart (*Figure 1H*, left), quadrants 1 (top right) and 3 (bottom left), respectively, displayed unsusceptible and susceptible mice in both Y-maze and MWM tests, while quadrants 2 (top left) and 4 (bottom right) displayed mice unsusceptible in one test but susceptible in the other one. The bar graph (*Figure 1H*, right) showed that the majority of CCI-Chronic mice exhibited consistent memory performance in the two behavioral tests (susceptible: 47%; unsusceptible: 43%), and only 10% of mice displayed susceptibility in a single test (Y-maze: 8%; MWM, 2%), suggesting good agreement between the two different assays. Given that the MWM test cannot be conducted once a week for multiple repeated measurements for the same batch of mice, the Y-maze test was used to investigate the duration of memory impairment induced by chronic pain. The results showed that memory impairment can last at least 63 days post CCI surgery, providing a workable time window for further investigations (*Figure 1—figure supplement 2A, B*).

We next examined whether the susceptibility or insusceptibility to memory impairment in mice with chronic pain is associated with the pain threshold. Linear regression, followed by a goodness-of-fit

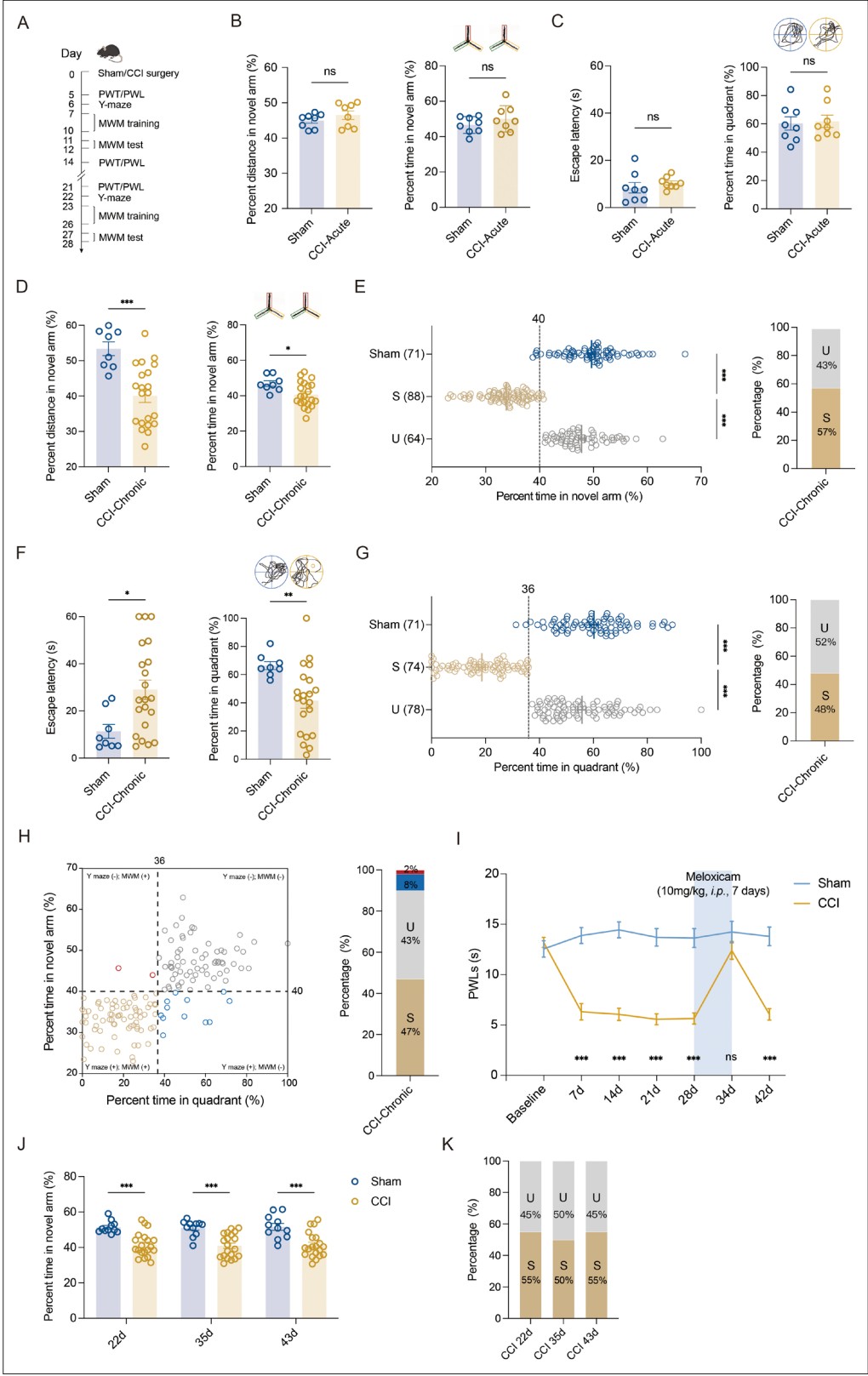

**Figure 1.** Segregation of mice with chronic pain into susceptible and unsusceptible subpopulations to memory impairment. (**A**) Timeline of CCI surgery, pain threshold tests, Y-maze test, and MWM training. (**B**) Representative traveling traces and statistical results of Y-maze test showing distance and percent time in the novel arm (red) in CCI-Acute mice (6d post CCI, *n* = 8–21). (**C**) Representative traveling traces and statistical results of MWM

*Figure 1 continued on next page*

*Figure 1 continued*

training showing escape latency and percent time in the quadrant in CCI-Acute mice (11d post CCI, *n* = 8–21). (**D**) Representative traveling traces and statistical results of Y-maze test showing distance and percent time in the novel arm (red) in CCI-Chronic mice (22d post CCI, *n* = 8–21). (**E**) Horizontal scatterplot depicting the distribution of ratio of time in novel arm for Sham, susceptible (S), and unsusceptible (U) mice in Y-maze test. Bar graph represents the ratio of S and U mice in CCI-Chronic mice (22d post CCI, *n* = 71–152). (**F**) Representative traveling traces and statistical results of MWM training showing escape latency and percent time in the quadrant in CCI-Chronic mice (27d post CCI, *n* = 8–21). (**G**) Horizontal scatterplot depicting the distribution of ratio of time in the quadrant for Sham, S, and U mice in MWM training. Bar graph represents the ratio of S and U mice in CCI-Chronic mice (27d post CCI, *n* = 71–152). (**H**) Time in novel arm versus percent time in quadrant, for 152 CCI-Chronic mice. Each dot corresponds to one mouse. Colors of dots correspond to the groups of U in Y-maze test but S in MWM (red), U in both Y-maze and MWM (gray), S in Y-maze test but U in MWM (blue), and S in both Y-maze and MWM (yellow), respectively. Bar graph represents the ratio of each group in CCI-Chronic mice (*n* = 152). (**I**) PWLs before and after administration of meloxicam (10 mg/kg, *i.p.*, *n* = 10–20). (**J**) Performance of CCI mice in Y-maze test before and after meloxicam administration (10mg/kg, *i.p.*, *n* = 10–20). (**K**) Bar graph represents the ratio of U and S on 22d, 36d, and 43d after CCI (*n* = 10–20). Data were analyzed by unpaired *t* test or two-way analysis of variance (two-way ANOVA), followed by post hoc Tukey's multiple comparisons between multiple groups when appropriate. All data are presented as the mean ± SEM. ns, not significant; *p < 0.05; **p < 0.01; ***p < 0.001. CCI, chronic constrictive injury; d, day; MWM, Morris water maze; PWL, paw withdrawal latency; PWT, paw withdrawal threshold; U, unsusceptible; S, susceptible.

The online version of this article includes the following figure supplement(s) for figure 1:

**Figure supplement 1.** Behavioral assays of nociception and locomotor activity.

**Figure supplement 2.** Chronic pain-induced memory impairment lasts at least to 63d after CCI.

**Figure supplement 3.** Susceptibility or insusceptibility to chronic pain-induced memory impairment is irrelevant to pain threshold of chronic constriction injury (CCI)-treated mice.

**Figure supplement 4.** Analgesic effects of meloxicam on neuropathic pain of CCI.

measure of *R*-squared ($r^2$), was used to determine the correlation between the two variables: percent time spent in the novel arm and pain threshold, or percent time in the target quadrant and pain threshold. The statistics revealed no correlation between memory performance for each mouse against its thermal pain threshold ($r^2$ = 0.04, *Figure 1—figure supplement 3A*; $r^2$ = 0.05, *Figure 1—figure supplement 3B*). Furthermore, we administered the CCI-Chronic mice with the selective COX-2 inhibitor NSAID analgesic meloxicam. Recommended doses for meloxicam in mice range from 1 to 10 mg/kg *i.p.* (*Gaertner et al., 2008*), and the duration of action (10 mg/kg) lasts at least 24 hr on day 7 post intraperitoneal injection once daily (*Figure 1—figure supplement 4*). Here, CCI-Chronic mice were subjected to pain treatment by meloxicam (10 mg/kg) for 7 days (once daily from day 28 to 34 post CCI) (*Figure 1I*). During the duration of analgesic effects (day 34–35 post CCI) (*Figure 1I*), we conducted the Y-maze test and found that the analgesia could not relieve the memory impairment (*Figure 1J*). Consistently, the percentage of susceptible and unsusceptible mice remained stable (*Figure 1K*), further indicating that the susceptibility or insusceptibility to memory impairment is marginally correlated with the pain tolerance of mice.

## S1PR1 expression is decreased in the hippocampal DG of susceptible mice

Hippocampal DG plays a vital role in learning and memory formation. To identify molecular mechanisms possibly contributing to susceptibility to memory impairment, we analyzed the hippocampal DG by RNA-Seq on day 28 after CCI when the mice with chronic pain were segregated into unsusceptible and susceptible subpopulations by Y-maze and MWM tests. We detected a total of 510 differentially expressed genes between Sham and susceptible mice, with 330 genes (65%) upregulated and 180 genes (35%) downregulated (*Figure 2A*). We noticed that Kyoto Encyclopedia of Genes and Genomes (KEGG) analysis revealed significant enrichment of six downregulated genes in the lipid metabolism pathway for Sham versus susceptible mice (*Figure 2B, C*). Combining the trend analysis of Sham versus unsusceptible versus susceptible mice, we found in the trend pattern of *Figure 2D* (left: Sham; middle, unsusceptible; right: susceptible), sphingolipid metabolism was significantly enriched (*Figure 2E*). In particular, transcription of *Sptlc3* was downregulated (*Figure 2C, E*). *Sptlc3* encodes

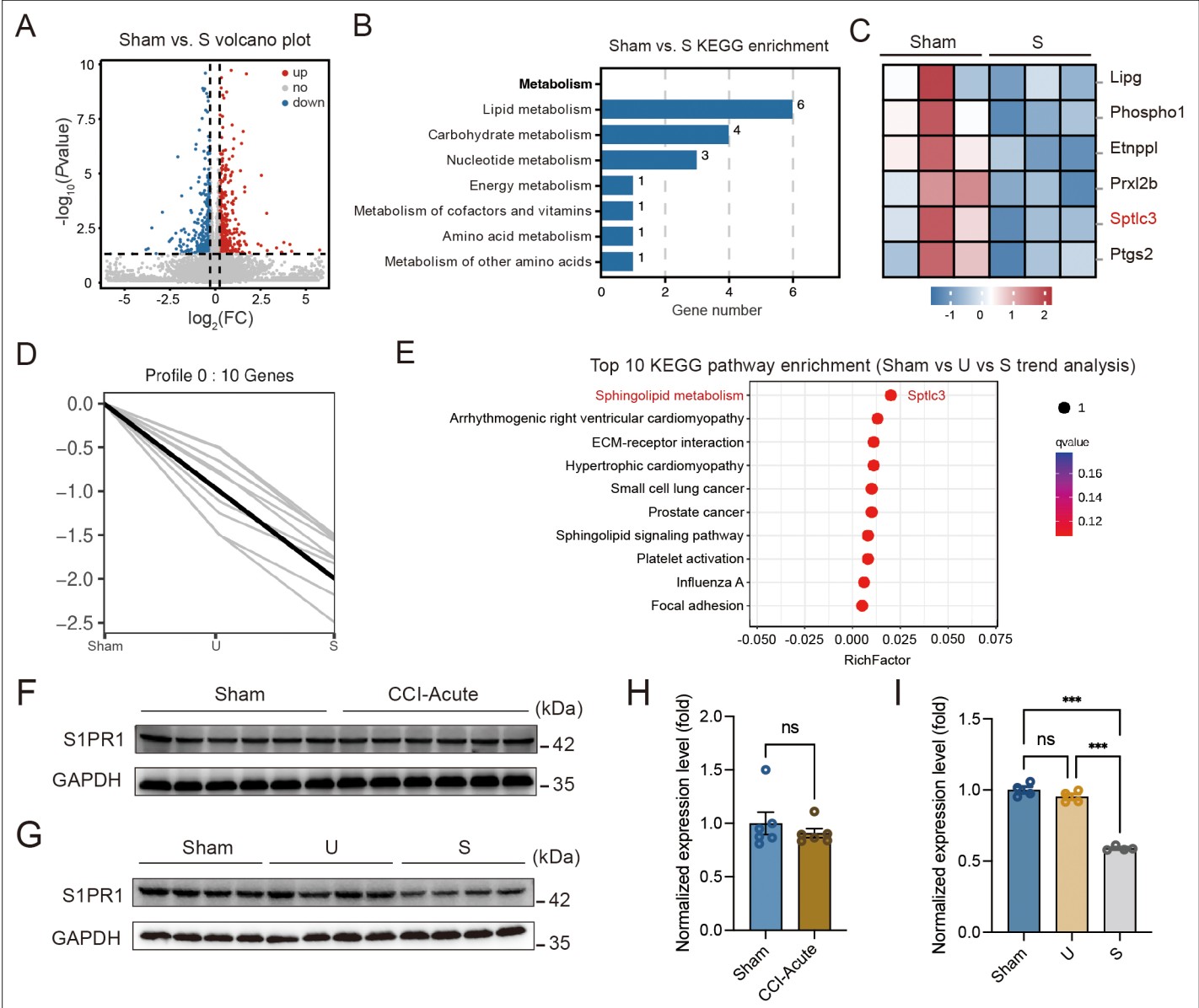

**Figure 2.** S1PR1 expression is decreased in the hippocampal dentate gyrus (DG) of susceptible mice. (**A**) Volcano plot showing RNA-Seq data for DG from Sham versus susceptible mice. Differentially expressed genes (DEGs) are designated in red (upregulation [up]) and blue (downregulation [down]) and defined as having an FDR of less than 0.05. (**B**) Bar plot showing significant enrichment of DEGs in various pathways related with metabolism for Sham versus susceptible mice. (**C**) Relative expression levels are shown for genes related with lipid metabolism upon susceptible as compared with Sham. (**D**) Trend pattern used for analysis of Sham versus U versus S. (**E**) Bubble diagram represents the top 10 enrichment of KEGG pathways. (**F–I**) Example western bands (**F**) and densitometric comparison (**H**) of the average expression of S1PR1 in DG lysates from Sham and CCI-Acute mice (7d post CCI). Lanes 1–6 represent Sham, Lanes 7–12 represent CCI-Acute ($n = 6$); example western bands (**G**) and densitometric comparison (**I**) of the average expression of S1PR1 in DG lysates from Sham, U, and S mice. Lanes 1–4 represent Sham, Lanes 5–8 represent U, and Lanes 9–12 represent S ($n = 4$). Data were analyzed by unpaired $t$ test or one-way analysis of variance (one-way ANOVA), followed by post hoc Tukey's multiple comparisons between multiple groups when appropriate. All data are presented as the mean ± SEM. ns, not significant; ***$p < 0.001$. CCI, chronic constrictive injury; d, day; U, unsusceptible; S, susceptible.

The online version of this article includes the following source data and figure supplement(s) for figure 2:

**Source data 1.** PDF file containing original western blots for **Figure 2F, G**, indicating the relevant bands.

**Source data 2.** Original files for western blot analysis displayed in **Figure 2F, G**.

**Figure supplement 1.** Characterization of expression profile of S1PR1 in the dentate gyrus (DG).

the subunit of the serine palmitoyltransferase (SPT) which catalyzes the rate-limiting step in sphingo-lipid biosynthesis (*Hornemann et al., 2009*). Defective SPT leads to disturbed sphingolipid homeo-stasis and failure of the subsequent production of metabolites such as glucosylceramide (*Zhang et al., 2011*) and sphingosine 1-phosphate (S1P) (*Gorshkova et al., 2012*), contributing to the occurrence of biological disorders. Previous studies have found that S1P/S1PR1 signaling is highly involved in hippocampus-engaged behaviors (*Yang et al., 2020*), we then verified the expression of S1PR1 in the hippocampal DG by western blotting (WB), with no change of S1PR1 observed on day 7 after CCI (*Figure 2F, H*) but significant downregulation in susceptible mice compared with Sham and unsuscep-tible ones (*Figure 2G, I*).

Furthermore, we characterized the expression profile of S1PR1 in the hippocampal DG. Immu-nofluorescence staining results showed that S1PR1 was mostly co-expressed with neuronal nuclear protein (NeuN) in neurons and merely co-expressed with glial fibrillary acidic protein (GFAP) in astro-cytes or ionized calcium-binding adaptor molecule 1 (Iba1) in microglia (*Figure 2—figure supplement 1A, C*). We also detected S1PR1 was highly co-expressed with calcium-calmodulin (CaM)-dependent protein kinase II (CaMKII)-expressing excitatory neurons but sparsely with glutamic acid decarboxy-lase 67 (GAD67)-expressing inhibitory neurons (*Figure 2—figure supplement 1B, D*). Taken together, these findings raised the possibility that S1P/S1PR1 may participate in the occurrence of chronic pain-related memory impairment.

## Knockdown of S1PR1 in the hippocampal DG promotes memory impairment susceptibility

To figure out the causal link between S1PR1 and chronic pain-related memory impairment, we gener-ated recombinant adeno-associated virus 2/9 (AAV2/9) expressing a small hairpin RNA targeting *S1pr1* (rAAV-CaMKIIa-EGFP-5'miR-30a-shRNA(*S1pr1*)-3'-miR30a-WPREs, the shRNA sequence is provided in *Supplementary file 1a*). Following the schematic experimental procedure shown in *Figure 3A*, intra-DG injection was conducted (*Figure 3B* and *Figure 3—figure supplement 1*). We first confirmed the knockdown efficiency of the virus in the hippocampal DG using WB (*Figure 3C*). We then exam-ined the effects of knockdown of S1PR1 in the DG on pain threshold. Compared with mice expressing scramble shRNA, mice expressing sh*S1pr1* in the DG had no effects on pain sensation (*Figure 3D*). Next, we assessed the effects of loss of S1PR1 in the DG on memory-related behaviors. In the Y-maze test, reduction of S1PR1 worsened the performance of Sham-treated mice by reducing the distance traveled and time spent in the novel arm, as well as enhancing the memory impairment of CCI-treated mice (*Figure 3E*). The results were consistent in the MWM test (*Figure 3F*). Consequently, knockdown of S1PR1 in the CCI-treated mice led to more susceptible mice (up to 86%) to memory impairment (*Figure 3G*). Thus, the above results suggest that DG S1PR1 exerts a negative regulatory effect on chronic pain-related memory impairment.

## Upregulation of S1PR1 in the hippocampal DG prevents the development of memory impairment susceptibility

To gain a deeper understanding of the functional consequences of enhanced S1P/S1PR1 signaling in the hippocampal DG, we generated recombinant AAV2/9 expressing the S1PR1 coding sequence (rAAV-CaMKIIa-*S1pr1*-P2A-EGFP-WPRE-hGH-polyA, sequence referred to *Supplementary file 1a*). Following the experimental flowchart depicted in *Figure 4A*, intra-DG injection was conducted (*Figure 4B* and *Figure 4—figure supplement 1*). We first confirmed the overexpression efficiency of the virus in the hippocampal DG using WB (*Figure 4C*). Next, we assessed how S1PR1 overexpres-sion in the DG affected pain threshold. Compared with the mice expressing the scramble shRNA, mice overexpressing S1PR1 in the DG had no effects on pain sensitization (*Figure 4D*). We then examined whether overexpression of S1PR1 in the DG influences chronic pain-related memory impair-ment according to behavioral paradigms of Y-maze and MWM tests. In the Y-maze and MWM tests, overexpression of S1PR1 in CCI-treated mice significantly improved the spatial memory formation by promoting insusceptibility to memory impairment (up to 82%), but had no obvious effects on Sham-treated mice (*Figure 4E–G*).

A previous study demonstrated that chronic administration of the selective S1PR1 agonist SEW2871 for 14 days inhibited the reduction of S1PR1 expression and improved impaired spatial memory in rats (*Asle-Rousta et al., 2013*). The dose used in the study (*Asle-Rousta et al., 2013*; *Park et al.,*

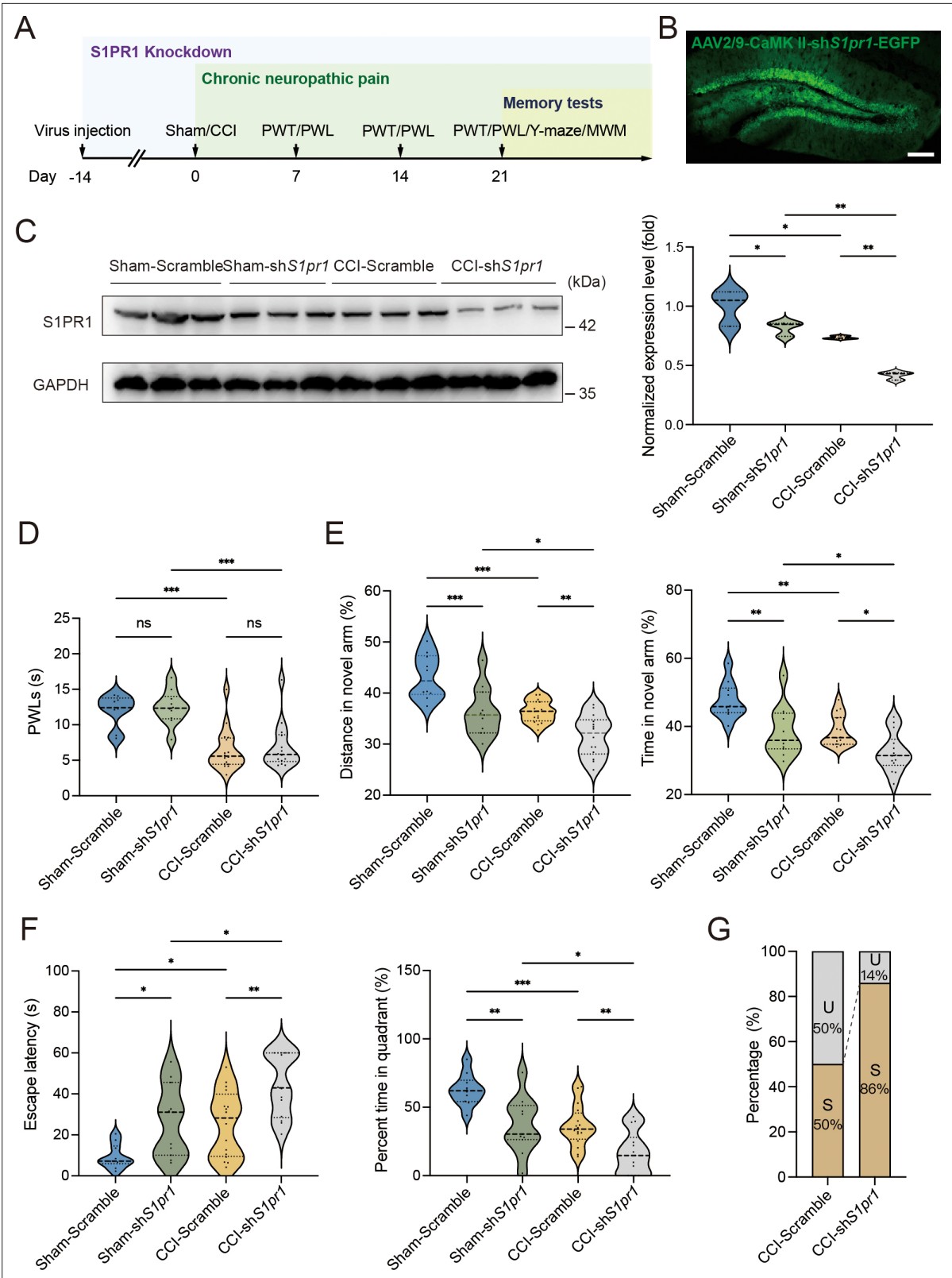

**Figure 3.** S1PR1 knockdown in the dentate gyrus (DG) induces memory impairment. (**A**) Timeline of intra-DG virus injection, CCI surgery, pain threshold tests, Y-maze test, and MWM training. (**B**) A confocal image showing virus expression in the DG (scale bar, 100 μm). (**C**) Example western bands showing efficient S1pr1 knockdown in the DG lysates from Sham-Scramble, Sham-sh*S1pr1*, CCI-Scramble, and CCI-sh*S1pr1*. Densitometric comparison of the average expression of S1PR1 (n = 6). (**D**) Pain threshold in Sham- and CCI-treated mice subjected to Scramble/sh*S1pr1* in the DG (n = 8–10). (**E**)

*Figure 3 continued on next page*

*Figure 3 continued*

Quantitative summary of Y-maze showing distances traveled and time spent in the novel arm in Sham- and CCI-treated mice subjected to Scramble/sh*S1pr1* in the DG (*n* = 10–16). (**F**) Quantitative summary of MWM training showing escape latency and time spent in the quadrant in Sham- and CCI-treated mice subjected to Scramble/sh*S1pr1* in the DG (*n* = 10–16). (**G**) Ratio of U and S in CCI-Scramble and CCI-sh*S1pr1* mice. Data were analyzed by two-way analysis of variance (two-way ANOVA), followed by post hoc Tukey's multiple comparisons between multiple groups. All data are presented as the mean ± SEM. ns, not significant; *$p < 0.05$; **$p < 0.01$; ***$p < 0.001$. CCI, chronic constrictive injury; MWM, Morris water maze; PWL, paw withdrawal latency; PWT, paw withdrawal threshold; U, unsusceptible; S, susceptible.

The online version of this article includes the following source data and figure supplement(s) for figure 3:

**Source data 1.** PDF file containing original western blots for *Figure 3C*, indicating the relevant bands.

**Source data 2.** Original files for western blot analysis displayed in *Figure 3C*.

**Figure supplement 1.** Zoomed-out images of the brain to show the precision of the virus injection (scale bar, 1000 μm).

*2010*) was effective to induce biological responses without causing kidney and liver injuries (*Park et al., 2010*). We then investigated whether a continuous 14-day local infusion of SEW2871 in the DG from day 7 post CCI could inhibit the reduction of S1PR1 expression and confer insusceptibility to memory impairment. For this purpose, a cannula was implanted into the DG of mice, and the S1PR1 agonist SEW2871 at a dose of 0.7 mg/kg/day (*Asle-Rousta et al., 2013*) was administered into the DG from day 7 to 21 post CCI surgery (*Figure 5A, B*). WB analysis revealed that bilateral DG injection of S1PR1 agonist SEW2871 inhibited the reduction of S1PR1 expression (*Figure 5C*), but did not alter the basal nociception of thermal stimuli (*Figure 5D*). In contrast, intra-DG administration of SEW2871 in CCI-treated mice significantly increased the distance traveled and time in the novel arm in the Y-maze (*Figure 5E*), decreased escape latency and increased percent time in the quadrant in the MWM (*Figure 5F*), resulting in an increased ratio (up to 80%) of unsusceptible mice (*Figure 5G*). Thus, it can be inferred from the above observations that upregulation of S1PR1 in the hippocampal DG promotes insusceptibility to chronic pain-related memory impairment.

## S1PR1 deficiency in the hippocampal DG modulates structural plasticity of DGCs

We next questioned how DG S1P/S1PR1 signaling modulates memory impairment. Adult neurogenesis enhances the plasticity of the hippocampus (*Zhang et al., 2008*), as well as preexisting granule neurons of the DG undergo dynamic alterations that include dendritic extension and retraction, synapse creation, and elimination (*Leuner and Gould, 2010*). Given that almost all the excitatory inputs from all sources toward DGCs are situated on their dendritic spines while the inhibitory connections are distributed in different layers (*Amaral et al., 2007*), we decided to examine the morphological changes of excitatory synapses which technically facilitates our observation. First, we utilized transmission electron microscope (TEM) to observe changes in the number of excitatory synapses and postsynaptic densities (PSD) in mice expressing sh*S1pr1* and scrambled shRNA (*Figure 6A*). In line with the behavioral tests, Sham-sh*S1pr1* mice showed a decreased number of excitatory synapses (*Figure 6B*), accompanied by a shorter PSD length and width compared with Sham-Scramble mice (*Figure 6C*). Furthermore, Golgi staining (*Figure 6D*) revealed that Sham-sh*S1pr1* mice had a lower dendritic intersection number (*Figure 6E*), shortened total dendritic length (*Figure 6F*), decreased mushroom/stubby type (*Figure 6G*, left) spines, and no change in thin/filopodia type (*Figure 6G*, right) dendritic spines. Additionally, CCI-sh*S1pr1* mice showed even more pronounced phenotypes as described above compared with CCI-Scramble mice (*Figure 6A–G*). Moreover, overexpression of S1PR1 in the DG significantly restored structural synaptic plasticity by increasing the number of excitatory synapses, PSD length and width (*Figure 7A–C*). Golgi staining results revealed that the intersection number of dendritic branches (*Figure 7D, E*), the total length of dendrites (*Figure 7F*), and the number of mushroom/stubby type dendritic spines (*Figure 7G*, left) increased in response to the intervention. Consistently, TEM images and Golgi staining revealed that continuous activation of S1PR1 in the DG significantly prevented the occurrence of defective synaptic plasticity (*Figure 7H–N*), indicating that activation of S1PR1 in the DG confers insusceptibility to memory impairment.

Next, we assessed whether the above structural plasticity changes can be observed in susceptible mice. First, we examined the number of neurons between susceptible and unsusceptible mice by staining for NeuN. The statistical result showed that the number is not grossly different between the

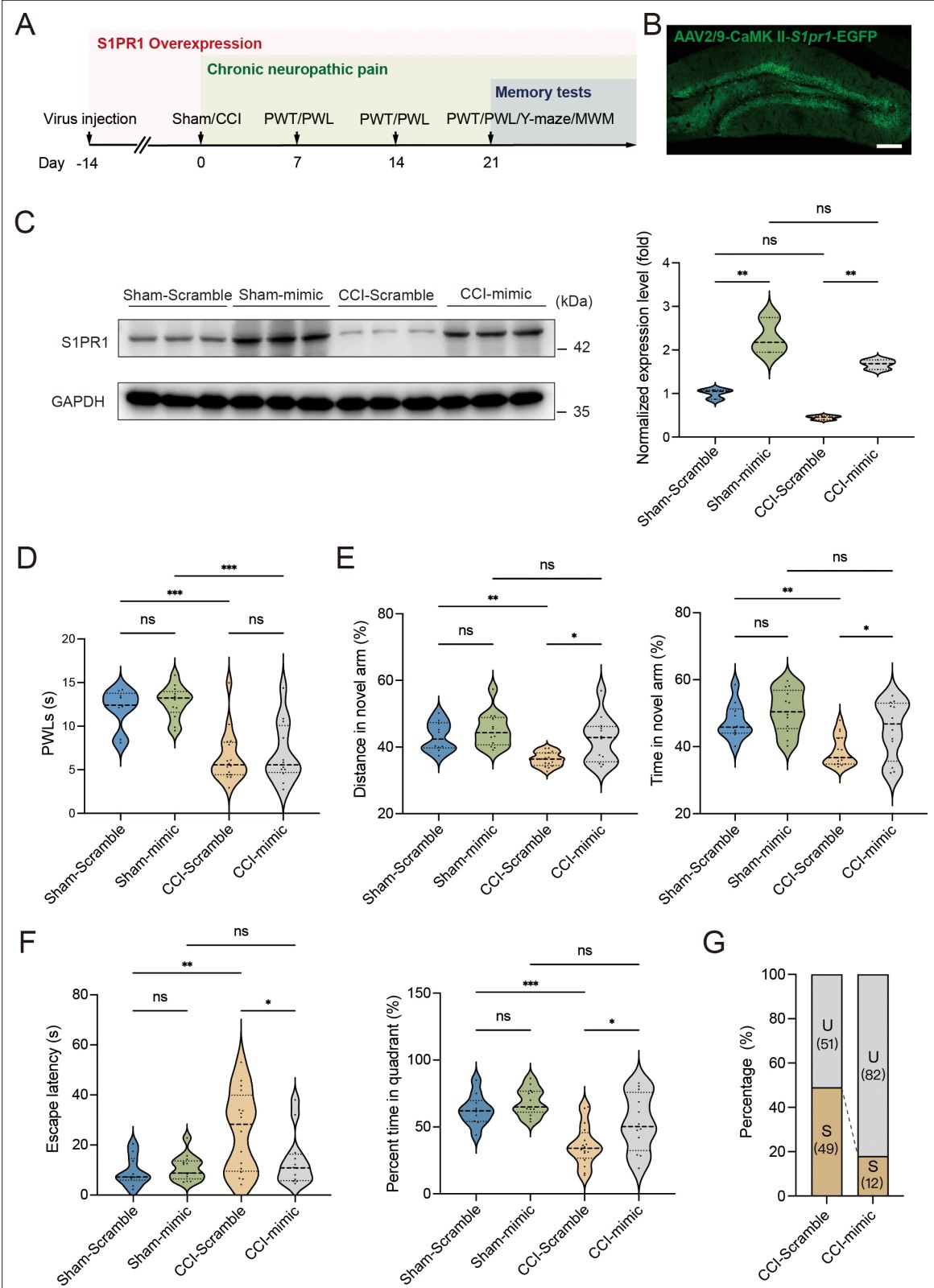

**Figure 4.** Overexpression of S1PR1 in the dentate gyrus (DG) rescues chronic pain-induced memory impairment. (**A**) Timeline of intra-DG virus injection, CCI surgery, pain threshold tests, Y-maze test, and MWM training. (**B**) A confocal image showing virus expression in the DG (scale bar, 100 µm). (**C**) Example western bands showing efficient S1PR1 overexpression in the DG lysates from Sham-Scramble, Sham-mimic, CCI-Scramble, and CCI-mimic. Densitometric comparison of the average expression of S1PR1 ($n = 6$). (**D**) Pain threshold in Sham- and CCI-treated mice subjected to Scramble/mimic

*Figure 4 continued on next page*

*Figure 4 continued*

in the DG (*n* = 8–10). (**E**) Quantitative summary of Y-maze showing distances traveled and time spent in the novel arm in Sham- and CCI-treated mice subjected to Scramble/mimic in the DG (*n* = 10–16). (**F**) Quantitative summary of MWM training showing escape latency and time spent in the quadrant in Sham- and CCI-treated mice subjected to Scramble/mimic in the DG (*n* = 10–16). (**G**) Ratio of U and S in CCI-Scramble and CCI-mimic mice. Data were analyzed by two-way analysis of variance (two-way ANOVA), followed by post hoc Tukey's multiple comparisons between multiple groups. All data are presented as the mean ± SEM. ns, not significant; *p < 0.05; **p < 0.01; ***p < 0.001. CCI, chronic constrictive injury; MWM, Morris water maze; PWL, paw withdrawal latency; PWT, paw withdrawal threshold; U, unsusceptible; S, susceptible.

The online version of this article includes the following source data and figure supplement(s) for figure 4:

**Source data 1.** PDF file containing original western blots for *Figure 4C*, indicating the relevant bands.

**Source data 2.** Original files for western blot analysis displayed in *Figure 4C*.

**Figure supplement 1.** Zoomed-out images of the brain to show the precision of the virus injection (scale bar, 1000 μm).

two populations (*Figure 6—figure supplement 1*). We then found that the synaptic structural plasticity in the hippocampal DG is disrupted in susceptible mice, but not in Sham and unsusceptible ones. TEM was also used to observe changes in the number of excitatory synapses and PSD (*Figure 6—figure supplement 2A, H*). The results showed that 7-day CCI mice exhibited a similar number of excitatory synapses (*Figure 6—figure supplement 2B*), as well as PSD length and width, to Sham mice (*Figure 6—figure supplement 2C*). On day 21 post CCI, susceptible mice displayed a decreased number of excitatory synapses (*Figure 6—figure supplement 2I*), accompanied by shortened PSD length and width compared with unsusceptible and Sham mice (*Figure 6—figure supplement 2J*). Furthermore, Golgi staining was utilized to identify the dendritic formation and spine morphology (*Figure 6—figure supplement 2D, K*). Seven-day CCI mice showed no altered dendritic intersection number (*Figure 6—figure supplement 2E*), total dendritic length (*Figure 6—figure supplement 2F*), mushroom/stubby type (*Figure 6—figure supplement 2G*, left), and thin/filopodia type (*Figure 6—figure supplement 2G*, right) dendritic spines. On day 21 post CCI, susceptible mice displayed a decreased dendritic intersection number (*Figure 6—figure supplement 2L*), shortened total dendritic length (*Figure 6—figure supplement 2M*), and a decreased number of mushroom/stubby type dendritic spines (*Figure 6—figure supplement 2N*, left) compared with unsusceptible and Sham mice. However, there were no changes in the number of thin/filopodia type (*Figure 6—figure supplement 2N*, right) dendritic spines. Overall, these findings suggest that the variation in structural plasticity of the hippocampal DG underlies susceptibility and insusceptibility to chronic pain-related memory impairment, and it is modulated by S1P/S1PR1 signaling.

## Defective S1P/S1PR1 signaling induced dysregulation of actin cytoskeleton organization in susceptible mice

Neuronal structural plasticity, such as morphogenesis of dendrites and dendritic spines, is primarily regulated by the actin cytoskeleton (*Luo, 2002*; *Hotulainen and Hoogenraad, 2010*). Disruption of normal actin organization has been associated with numerous neurological and psychiatric diseases (*Bernstein et al., 2011*). It has been demonstrated that S1P/S1PRs modulate significant cytoskeletal rearrangements in various cellular systems through actin regulatory proteins such as Rho GTPases, including RAC1 and CDC42 (*Cui et al., 2022*; *Donati and Bruni, 2006*). These proteins promote filopodia formation by stimulating actin polymerization through WAVE and the ARP2/3 complex (*Donati and Bruni, 2006*; *Green and Cyster, 2012*). Additionally, we analyzed the RNA-Seq data to identify genes that may be involved in S1PR1-regulated cytoskeletal dynamics in our animal model. In the KEGG analysis comparing Sham versus susceptible mice and the trend analysis (Sham versus unsusceptible versus susceptible mice), integrin α2 (encoded by Itga2) was found to be enriched and downregulated among cell adhesion molecules and regulation of actin cytoskeleton pathway genes (*Figure 8A, B*). Integrins serve as linkers between the extracellular matrix and intracellular actin cytoskeleton, mediating cytoskeletal organization (*Adorno-Cruz and Liu, 2019*). Therefore, we asked whether S1P/S1PR1 induces structural plasticity in susceptible mice by regulating actin dynamics in the DG. To address this, we verified the expression levels of actin regulatory proteins, including RAC1, CDC42, ARP2, ARP3, and ITGA2, in 7-day CCI mice (*Figure 8C, D*), unsusceptible mice, and susceptible mice (*Figure 8E, F*). To figure out the regulation is specifically S1PR1-dependent, but not through other S1PRs and Rho GTPase, we also checked the expression level of S1PR2 and Rho GTPase RhoA

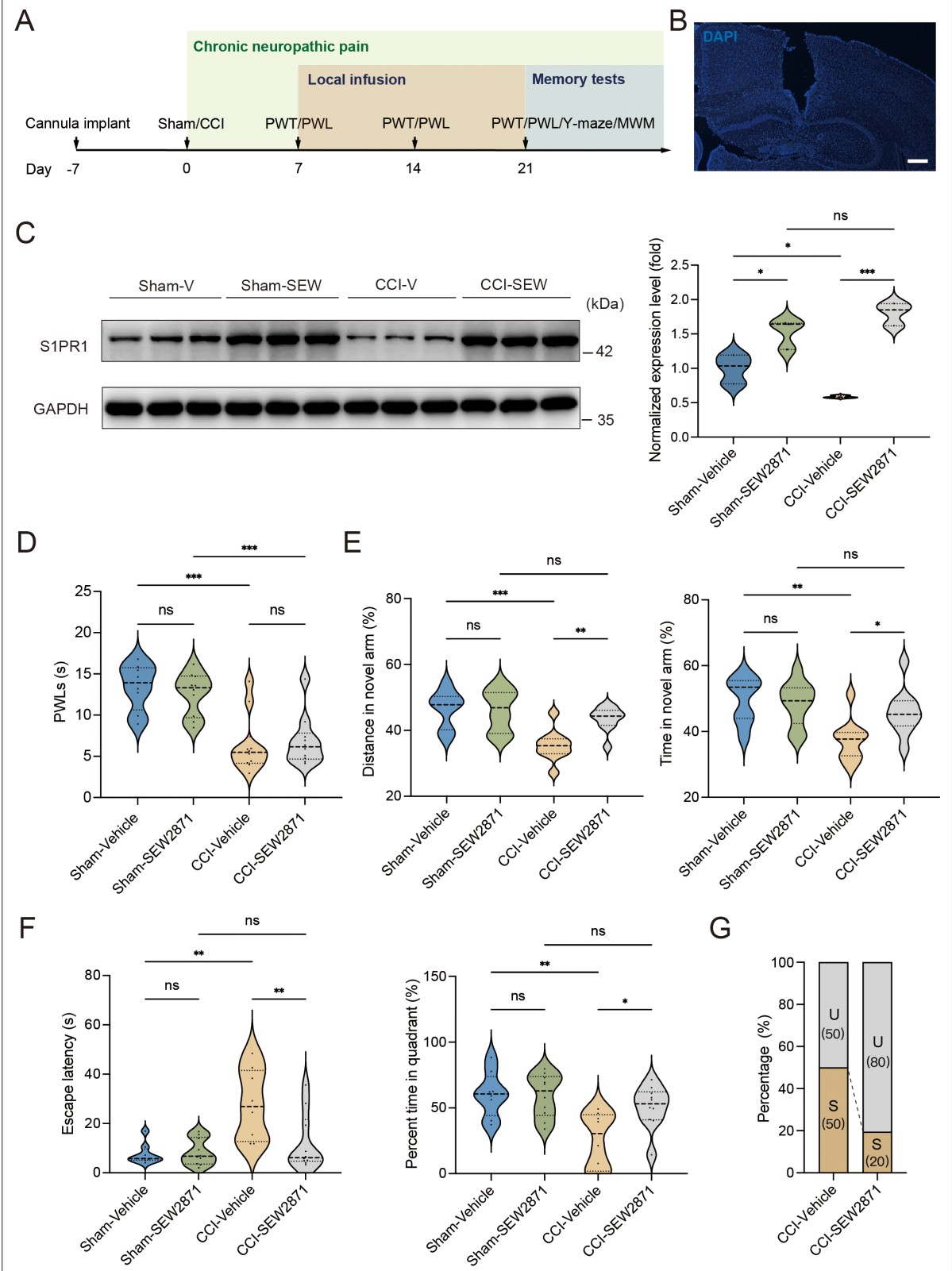

**Figure 5.** Local infusion of SEW2871 in the dentate gyrus (DG) prevented the presence of chronic pain-induced memory impairment. (**A**) Timeline of cannula implant, CCI surgery, pain threshold tests, Y-maze test, and Morris water maze (MWM) training. (**B**) Confocal image showing cannula implanted in the DG (scale bar, 100 μm). (**C**) Example western bands showing expression of S1PR1 in DG lysates from Sham-Vehicle, Sham-SEW2871, CCI-Vehicle, and CCI-SEW2871. Densitometric comparison of the average expression of S1PR1 ($n = 6$). (**D**) Pain threshold in Sham- and CCI-treated mice subjected

*Figure 5 continued on next page*

*Figure 5 continued*

to local infusion of vehicle/SEW2871 in the DG ($n$ = 8–10). (**E**) Quantitative summary of Y-maze showing distances traveled and time spent in the novel arm in Sham- and CCI-treated mice subjected to local infusion of vehicle/SEW2871 in the DG ($n$ = 10–16). (**F**) Quantitative summary of MWM training showing escape latency and time spent in the quadrant in Sham- and CCI-treated mice subjected to local infusion of vehicle/SEW2871 in the DG ($n$ = 10–16). (**G**) Ratio of U and S in CCI-vehicle and CCI-SEW2871 mice. Data were analyzed by two-way analysis of variance (two-way ANOVA), followed by post hoc Tukey's multiple comparisons between multiple groups. All data are presented as the mean ± SEM. ns, not significant; *$p$ < 0.05; **$p$ < 0.01; ***$p$ < 0.001. CCI, chronic constrictive injury; MWM, Morris water maze; PWL, paw withdrawal latency; PWT, paw withdrawal threshold; DG, dentate gyrus; U, unsusceptible; S, susceptible.

The online version of this article includes the following source data for figure 5:

**Source data 1.** PDF file containing original western blots for *Figure 5C*, indicating the relevant bands.

**Source data 2.** Original files for western blot analysis displayed in *Figure 5C*.

which interacts with S1PR2 (*Cui et al., 2022*; *Donati and Bruni, 2006*; *Reinhard, 2017*; *Figure 8— figure supplement 1*). We observed a significant decrease in the levels of RAC1, CDC42, ARP2, ARP3, and ITGA2 in susceptible mice, but not in Sham and unsusceptible mice. As expected, there is no change in the expression level of S1PR2 and RhoA in CCI animals. To determine whether the reduction in levels of these proteins is associated with downregulated S1PR1 signaling, we examined lysates of hippocampal DG tissue extracted from S1PR1 DG conditional knockdown mice, revealing decreased levels of RAC1, CDC42, ARP2, ARP3, and ITGA2 as assayed by WB (*Figure 8G, H*). Conversely, S1PR1 overexpression in the DG led to increased levels of RAC1, CDC42, ARP2, ARP3, and ITGA2 (*Figure 8I, J*). These data indicate that S1pr1 functions upstream of Rac1, Cdc42, Arp2/3, and Itga2. Given a previous study suggesting the involvement of an isoform of integrin ITGB4 in S1pr1-mediated Rac1 activation (*Ephstein et al., 2013*), our results raise the possibility that dysregulated actin rearrangement elicited by S1PR1 downregulation in the DG may be influenced by Itga2-dependent activation of the Rac1/Cdc42 signaling cascade and Arp2/3-dependent actin polymerization.

## S1P/S1PR1 regulates actin polymerization by interaction with ITGA2

We then investigated the regulatory role of S1P/S1PR1 signaling in actin polymerization. The polymerization of monomeric actin (G-actin) into actin filaments (F-actin) to form the actin cytoskeleton frequently occurs primarily at or near the plasma membrane. The organization of actin filaments determines the shape, stiffness, and movement of the cell surface and also facilitates spine morphology and function (*Cooper, 2000*). Thus, we first examined actin polymerization by quantifying the transition from G- to F-actin transition. Densitometric analysis of F-/G-actin western blots (*Figure 9A, B*) indicated a significant decrease in the relative percentage of F-actin in the DG of susceptible mice compared with that of Sham and unsusceptible mice. Immunofluorescence observations using a fluorescently conjugated phalloidin, which binds only to F-actin, revealed that HT-22 mouse hippocampal neuronal cells with knockdown of S1PR1 had accumulated F-actin aggregates mostly around the nuclei and lost the majority of the thinner filament bundles (*Figure 9C*) compared with control cells. Similarly, we utilized primary hippocampal cells to investigate the neuron morphology and distribution of F-actin. We observed less branches and accumulated F-actin aggregates rather than diffused distribution compared with control cells. Furthermore, we investigated whether S1PR1 regulates actin polymerization via interaction with ITGA2. We conducted yeast two-hybrid screening in vitro and a co-immunoprecipitation assay in vivo to examine the interaction between the two proteins. To identify putative interaction between S1PR1 and ITGA2, the full sequences of S1PR1 (cloned into the pBT3-STE vector) and ITGA2 (cloned into the pPR3-C vector) were used as bait and prey, respectively (*Figure 9—figure supplement 1A, B*). The auto-activation test showed that positive controls (the pNubG-Fe65 and pTSU2-APP vector together) grew on the DDO (SD/-Trp/-Leu), TDO (SD/-Trp/-Leu/-His), and QDO (SD/-Trp/-Leu/-His/-Ade) plates. Meanwhile, the negative control (the pPR3-N and pTSU2-APP vectors together), the pPR3-N empty vector, and the pBT3-STE vector with S1pr1 grew on the DDO and TDO but not on the QDO plate (*Figure 9—figure supplement 2*). These results indicated that S1PR1 did not exhibit auto-activation activity in yeast. Next, a yeast two-hybrid assay was performed by co-transforming pBT3-STE-S1pr1 and pPR3-C-Itga2 in the NMY51 yeast strain. Yeast cells harboring both S1pr1 and Itga2 grew vigorously on both DDO media and QDO/X-gal media (*Figure 9D*). Additionally, in a CoIP assay, total protein extracts from the mice DG were immunoprecipitated by the anti-S1PR1-specific antibody and analyzed by immunoblotting probed with

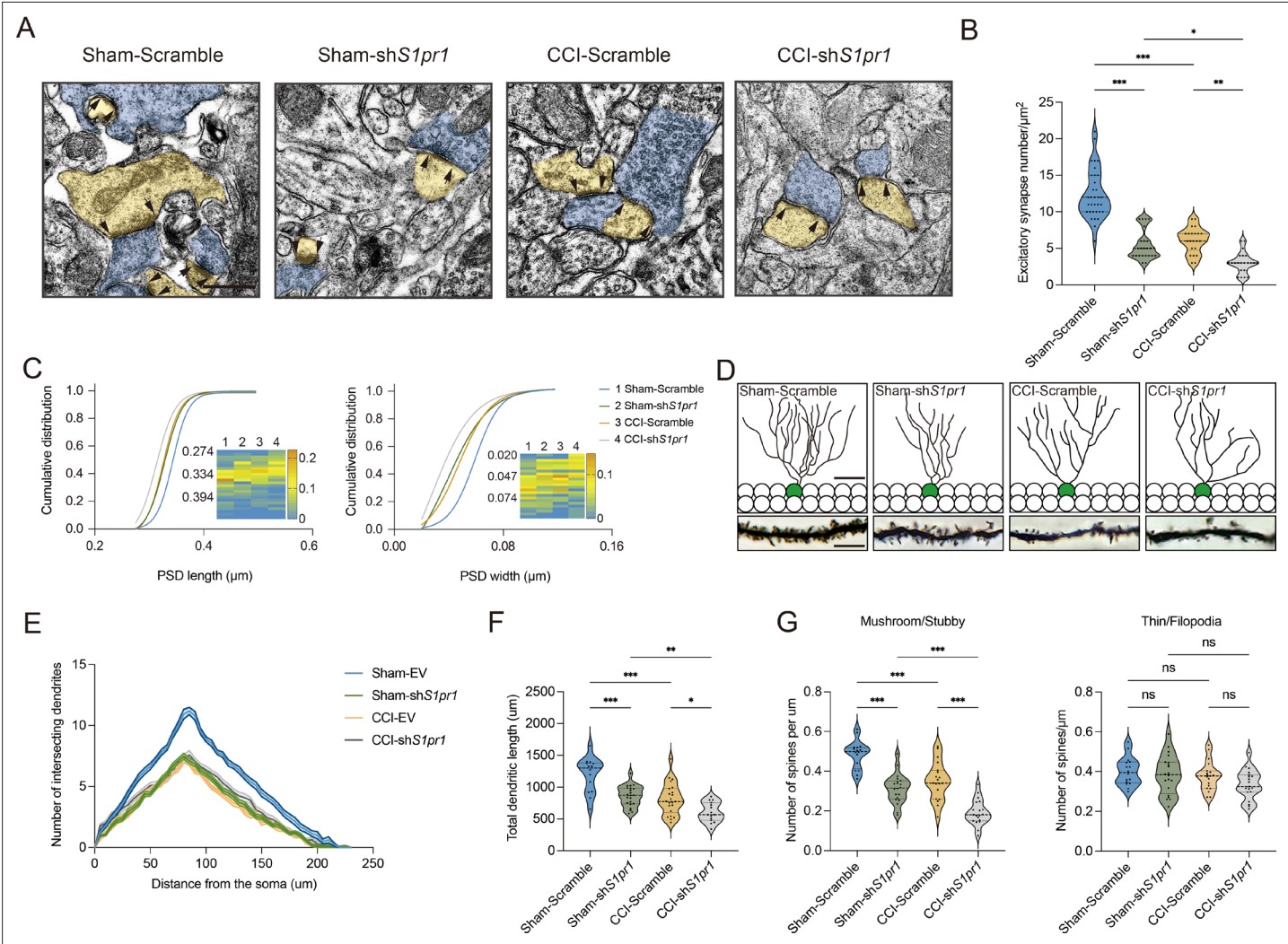

**Figure 6.** S1PR1 knockdown in the dentate gyrus (DG) impairs synaptic plasticity. (**A**) Representative TEM images of synapses in the DG in Sham- and CCI-treated mice subjected to Scramble/sh*S1pr1* in the DG. Blue indicates presynaptic site and yellow indicates postsynaptic sites of excitatory synapses, respectively (scale bar, 500 nm). (**B**) Mean number of excitatory synapses per μm² of DG in Sham- and CCI-treated mice subjected to Scramble/sh*S1pr1* (*n* = 18–24 from 4 mice/group). (**C**) Cumulative distribution plots for the lengths and widths of postsynaptic density in the DG in Sham- and CCI-treated mice subjected to Scramble/sh*S1pr1* in the DG (*n* = 121–162 from 4 mice/group). (**D**) Representative Golgi-staining images of dendritic spine morphology from the DG in Sham- and CCI-treated mice subjected to Scramble/sh*S1pr1* in the DG (scale bar, top: 50 μm; bottom:10 μm). (**E**) The number of intersections of all dendritic branches in Sham- and CCI-treated mice subjected to Scramble/sh*S1pr1* in the DG (*n* = 18–24 from 4 mice/group). (**F**) Violin plots indicate the total dendritic length (*n* = 18–24 from 4 mice/group). (**G**) Violin plots indicate the number of mushroom/stubby type dendritic spines (left), and the number of thin/filopodia type dendritic spines (right) in the DG of Sham- and CCI-treated mice subjected to Scramble/sh*S1pr1* in the DG (*n* = 18–24 from 4 mice/group). Data were analyzed by two-way analysis of variance (two-way ANOVA), followed by post hoc Tukey's multiple comparisons between multiple groups when appropriate. All data are presented as the mean ± SEM. ns, not significant; *p < 0.05; **p < 0.01; ***p < 0.001. CCI, chronic constrictive injury; TEM, transmission electron microscope.

The online version of this article includes the following figure supplement(s) for figure 6:

**Figure supplement 1.** The number of neurons in the hippocampal dentate gyrus.

**Figure supplement 2.** Susceptible mice exhibit altered excitatory synaptic plasticity in the hippocampal dentate gyrus.

the anti-S1PR1 and anti-ITGA2 antibodies with immunoglobulin G as the negative control. The in vivo CoIP assay showed that S1PR1 interacts with ITGA2 in the DG (*Figure 9E*). The above results indicated that S1PR1 may physically interact with ITGA2. To determine the functional interaction between S1PR1 and ITGA2 in regulating chronic pain-related memory impairment, a recombinant adeno-associated virus 2/9 (AAV2/9) expressing a small hairpin RNA targeting *Itga2* was generated (rAAV-CaMKIIa-EGFP-5'miR-30a-shRNA (*Itga2*)-3'-miR30a-WPREs, the sequence of shRNA referred

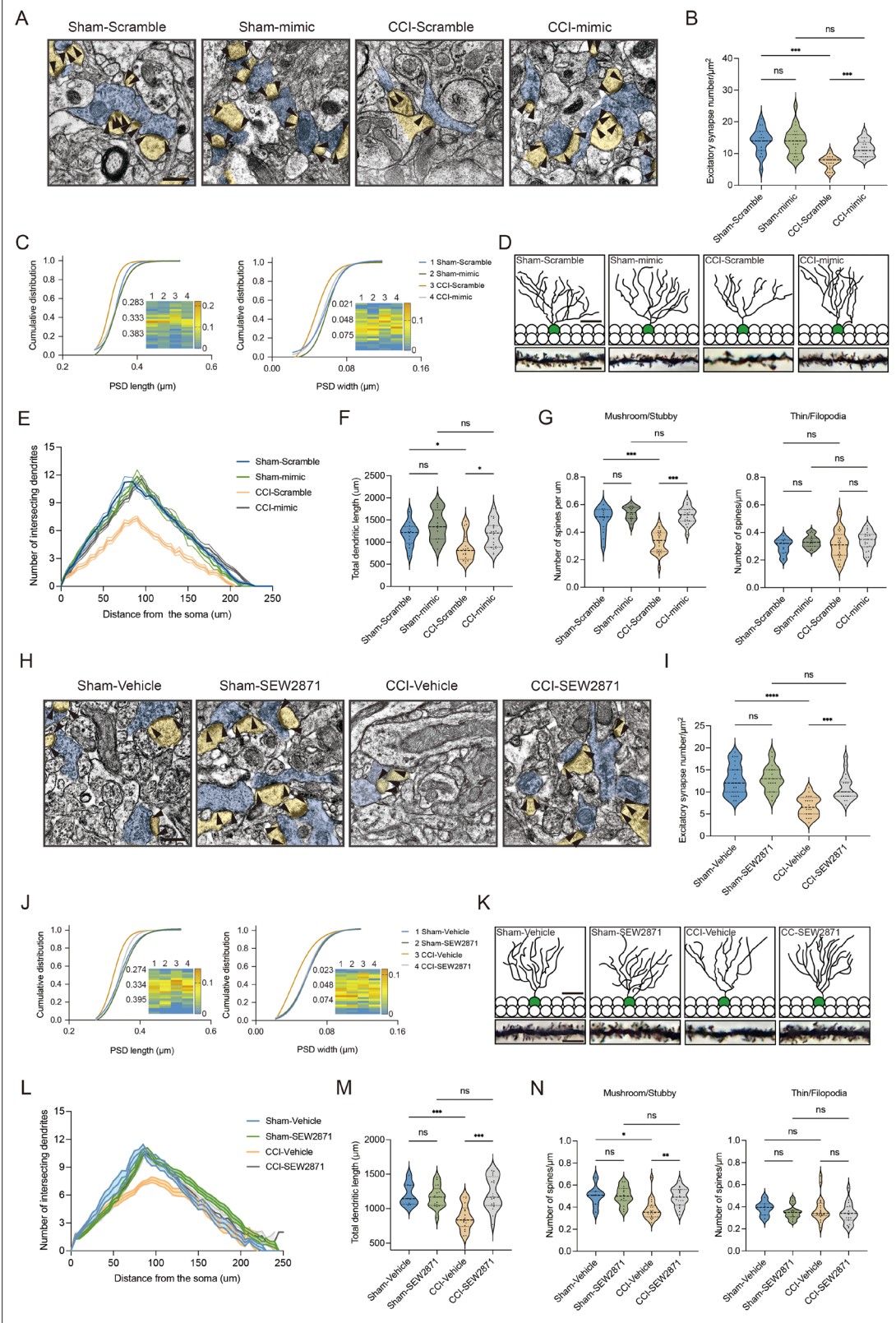

**Figure 7.** Overexpression of S1PR1 or local infusion of SEW2871 in the DG maintained the synaptic structural plasticity. (**A**) Representative TEM images of synapses in the DG in Sham- and CCI-treated mice subjected to Scramble/mimic in the DG. Blue indicates presynaptic site and yellow indicates postsynaptic sites of excitatory synapses, respectively. Synaptic densities are bracketed by arrows (scale bar, 500 nm). (**B**) Mean number of excitatory synapses per μm$^2$ of DG in Sham- and CCI-treated mice subjected to Scramble/mimic ($n$ = 18–24 from 4 mice/group). (**C**) Cumulative distribution plots

*Figure 7 continued on next page*

*Figure 7 continued*

for the lengths and widths of postsynaptic density in the DG in Sham- and CCI-treated mice subjected to Scramble/mimic in the DG ($n$ = 121–162 from 4 mice/group). (**D**) Representative Golgi-staining images of dendritic spine morphology from the DG in Sham- and CCI-treated mice subjected to Scramble/mimic in the DG (scale bar, top: 50 µm; bottom:10 µm). (**E**) The number of intersections of all dendritic branches in Sham- and CCI-treated mice subjected to Scramble/mimic in the DG ($n$ = 18–24 from 4 mice/group). (**F**) Violin plots indicate the total dendritic length ($n$ = 18–24 from 4 mice/group). (**G**) The number of mushroom/stubby type dendritic spines (left), and the number of thin/filopodia type dendritic spines (right) in the DG of Sham- and CCI-treated mice subjected to Scramble/mimic in the DG ($n$ = 18–24 from 4 mice/group). (**H**) Representative TEM images of synapses in the DG in Sham- and CCI-treated mice subjected to local infusion of vehicle/SEW2871 in the DG. Blue indicates presynaptic site and yellow indicates postsynaptic sites of excitatory synapses, respectively. Synaptic densities are bracketed by arrows (scale bar, 500 nm). (**I**) Mean number of excitatory synapses per µm$^2$ of DG in Sham- and CCI-treated mice subjected to local infusion of vehicle/SEW2871 ($n$ = 18–24 from 4 mice/group). (**J**) Cumulative distribution plots for the lengths and widths of postsynaptic density in the DG in Sham- and CCI-treated mice subjected to local infusion of vehicle/SEW2871 in the DG ($n$ = 121–162 from 4 mice/group). (**K**) Representative Golgi-staining images of dendritic spine morphology from the DG in Sham- and CCI-treated mice subjected to local infusion of vehicle/SEW2871 in the DG (scale bar, top: 50 µm; bottom:10 µm). (**L**) The number of intersections of all dendritic branches in Sham- and CCI-treated mice subjected to local infusion of vehicle/SEW2871 in the DG ($n$ = 18–24 from 4 mice/group). (**M**) Violin plots indicate the total dendritic length ($n$ = 18–24 from 4 mice/group). (**N**) The number of mushroom/stubby type dendritic spines (left), and the number of thin/filopodia type dendritic spines (right) in the DG of Sham- and CCI-treated mice subjected to local infusion of vehicle/SEW2871 in the DG ($n$ = 18–24 from 4 mice/group). Data were analyzed by two-way analysis of variance (two-way ANOVA), followed by post hoc Tukey's multiple comparisons between multiple groups. All data are presented as the mean ± SEM. ns, not significant; *$p < 0.05$; **$p < 0.01$; ***$p < 0.001$; ****$p < 0.0001$. CCI, chronic constrictive injury; DG, dentate gyrus; TEM, transmission electron microscope.

to *Supplementary file 1a*). Following the schematic experimental procedure shown in *Figure 9F*, intra-DG injection of the ITGA2 knockdown virus was conducted (*Figure 9G* and *Figure 9—figure supplement 3*). We first confirmed the knockdown efficiency of the virus in the hippocampal DG using WB (*Figure 9H*). We then examined the effects of knockdown of ITGA2 in the DG on pain threshold and memory performance. Consistent with the effects of S1PR1 knockdown in the DG, compared with mice expressing scramble shRNA, mice expressing sh*Itga2* in the DG had no effects on the pain sensation (*Figure 9I*) and worsened memory-related behaviors in the Y-maze and MWM tests (*Figure 9J, K*). To further demonstrate that S1PR1 and ITGA2 participate in the same pathway, we knocked down the two proteins at the same time (*Figure 9L, M* and *Figure 9—figure supplement 4*). As expected, it did not elicit addictive effects on behavioral tests of Y-maze and MWM tests compared to the knockdown of each one of them in isolation (*Figure 9N, O*). Overall, these findings suggest that DG S1PR1 may govern the susceptibility of memory impairment by regulating actin polymerization via interaction with ITGA2.

## Discussion

The current study aimed to explore a disease model of chronic pain-related memory impairment and uncover the molecular underpinnings of both susceptibility to chronic pain-related memory impairment and factors that promote susceptibility to such variations. We established a paradigm to segregate mice with chronic pain into memory impairment-susceptible and -unsusceptible subpopulations. Susceptible mice displayed long-lasting memory impairment 21 days after CCI surgery, while unsusceptible mice continued to maintain normal memory function. Importantly, TEM and Golgi staining assays revealed that susceptible mice exhibited signs of reduced excitatory synapse formation and abnormal spine morphology in the brain area of the hippocampal DG involved in cognition/memory. Interestingly, the phenotypic variability in mice is attributed to dysregulation of S1P/S1PR1/integrin α2 signaling-induced disorganization of actin cytoskeleton through the Rac1/Cdc42 signaling and Arp2/3 cascade (*Figure 10*). These are significant findings that demonstrate with certainty that comorbidity of memory impairment in complicated chronic pain syndromes has a pathophysiological substrate in the brain that could be a key therapeutic target for intervention.

Several studies have shown that neuronal plastic changes in the hippocampus are highly relevant to chronic pain-induced memory impairment. For example, the hippocampal extracellular matrix exhibits aberrated structural synaptic plasticity connected to deficiencies in working location memory in a mouse model of chronic pain. These deficits are also correlated with decreased hippocampus dendritic complexity (*Tajerian et al., 2018*). Moreover, rodents with neuropathic pain show altered short-term synaptic plasticity related to the decrease in hippocampus volume detected in patients. The anomaly may be the cause of the typical learning and emotional deficiencies seen in people with chronic pain

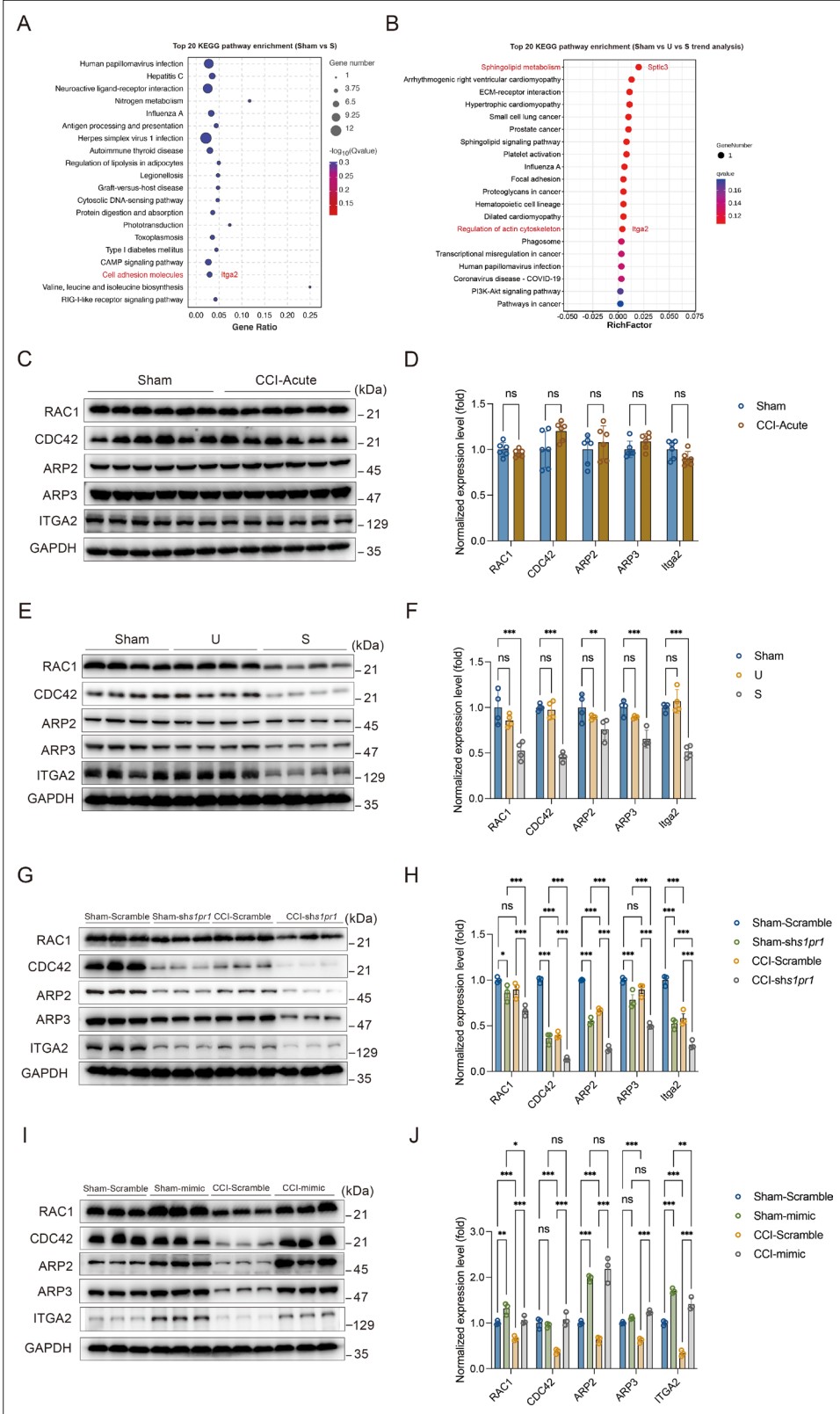

**Figure 8.** Susceptible mice, S1PR1-knockdown mice, and S1PR1 overexpression mice exhibit altered CDC42/RAC1 activity, ARP2/3-dependent actin signaling and engage ITGA2 in the dentate granule cells (DGCs). (**A**) Bubble diagram showing significant enrichment of differentially expressed genes (DEGs) in top 20 KEGG pathways for Sham versus susceptible mice. (**B**) Bubble diagram represents the top 20 enrichment of KEGG pathways using

*Figure 8 continued on next page*

*Figure 8 continued*

analysis of Sham versus U versus S. (**C**) Example western bands showing expression of RAC1, CDC42, ARP2, ARP3, and ITGA2 in DG lysates from Sham and CCI-Acute mice (7d post CCI). (**D**) Densitometric comparison of the average expression of RAC1, CDC42, ARP2, ARP3, and ITGA2 (*n* = 6). (**E**) Example western bands showing expression of RAC1, CDC42, ARP2, ARP3, and ITGA2 in DG lysates from Sham, U, and S mice. (**F**) Densitometric comparison of the average expression of RAC1, CDC42, ARP2, ARP3, and ITGA2 (*n* = 6). (**G**) Example western bands showing expression of RAC1, CDC42, ARP2, ARP3, and ITGA2 in DG lysates from in Sham- and CCI-treated mice subjected to Scramble/sh*S1pr1* in the DG. (**H**) Densitometric comparison of the average expression of RAC1, CDC42, ARP2, ARP3, and ITGA2 (*n* = 6). (**I**) Example western bands showing expression of RAC1, CDC42, ARP2, ARP3, and ITGA2 in DG lysates from in Sham- and CCI-treated mice subjected to Scramble/*S1pr1*-mimic in the DG. (**J**) Densitometric comparison of the average expression of RAC1, CDC42, ARP2, ARP3, and ITGA2 (*n* = 6). Data were analyzed by unpaired *t* test or two-way analysis of variance (two-way ANOVA), followed by post hoc Tukey's multiple comparisons between multiple groups when appropriate. All data are presented as the mean ± SEM. ns, not significant; *p < 0.05; **p < 0.01; ***p < 0.001. CCI, chronic constrictive injury; DG, dentate gyrus; U, unsusceptible; S, susceptible.

The online version of this article includes the following source data and figure supplement(s) for figure 8:

**Source data 1.** PDF file containing original western blots for *Figure 8C, E, G, I*, indicating the relevant bands.

**Source data 2.** Original files for western blot analysis displayed in *Figure 8C, E, G, I*.

**Figure supplement 1.** Expression levels of S1PR2 and RhoA in Sham and CCI animals.

**Figure supplement 1—source data 1.** PDF file containing original western blots for *Figure 8—figure supplement 1A*, indicating the relevant bands.

**Figure supplement 1—source data 2.** Original files for western blot analysis displayed in *Figure 8—figure supplement 1A*.

(*Mutso et al., 2012*). In addition, the hippocampus regions involved in the processing of pain information show abnormalities in neurite arborization, dendritic length, and dendritic spine architecture (*Tyrtyshnaia and Manzhulo, 2020*). Consistently, we found that hippocampal DG neurons displayed decreased excitatory synapses and altered spine morphology in susceptible mice. Although extensive investigations have proven the link between hippocampal DG function with memory formation, it remains unclear what regulates the plastic changes in the hippocampal DG, a well-studied brain area responsible for memory formation, and how it participates in the modulation of memory impairment in chronic pain. Interestingly, using RNA-Seq, we detected significant transcriptional downregulation of sphingolipid metabolism in the DG, which we verified at protein levels due to dysregulation of S1P/S1PR1 signaling. A previous study found that the expression of S1PR1 is upregulated in freshly formed DG cells, which is required for neurite arborization and horizontal-to-radial repositioning of these cells (*Yang et al., 2020*). They also raised a question of whether S1PR1 regulates mature DGC activity. In the current study, we mainly focused on S1P function in mature DGCs. We knocked down DG S1PR1 and found that the loss of S1PR1 in the DG induced more susceptible mice to memory impairment without affecting pain threshold. Additionally, we also overexpressed DG S1PR1 and noticed that it promoted mice to be unsusceptible to memory impairment, similarly irrelevant to sensitivity to thermal pain stimuli. These findings imply that S1PR1 in the DG may exclusively negatively regulate chronic pain-related memory impairment. Nevertheless, CNS S1PR1 activation has also been reported in conditions with cisplatin-induced cognitive impairment (*Squillace et al., 2022*), and peripheral administration of either agonist or functional S1PR1 antagonist can ameliorate spatial memory impairment (*Asle-Rousta et al., 2013*; *Zhang et al., 2020*). Herein, the mechanisms of action of S1PR1 signaling (agonism or antagonism) with regard to the memory performance remain controversial.

Furthermore, our study pointed out that S1PR1 activation in the DG may not be involved in the processing of pain. To date, a growing body of evidence has shown that activation of S1P axis at spinal cord triggers the occurrence of peripheral sensitization of pain. Activation of S1PR1 in astrocytes in the spinal cord contributes to neuropathic pain (*Chen et al., 2019*), and mice with astrocyte-specific alterations of S1PR1 in the spinal cord did not experience neuropathic pain (*Stockstill et al., 2018*). Additionally, elevated S1P levels at spinal cord injury sites attract macrophages and microglia, and their activation worsens the inflammatory response (*Kimura et al., 2007*). Moreover, a S1PR1 antagonist lessens neuropathic pain induced by spinal cord injury by inhibiting neuroinflammation and glial scar formation (*Yamazaki et al., 2020*). In contrast with the evidence available in the spinal cord,

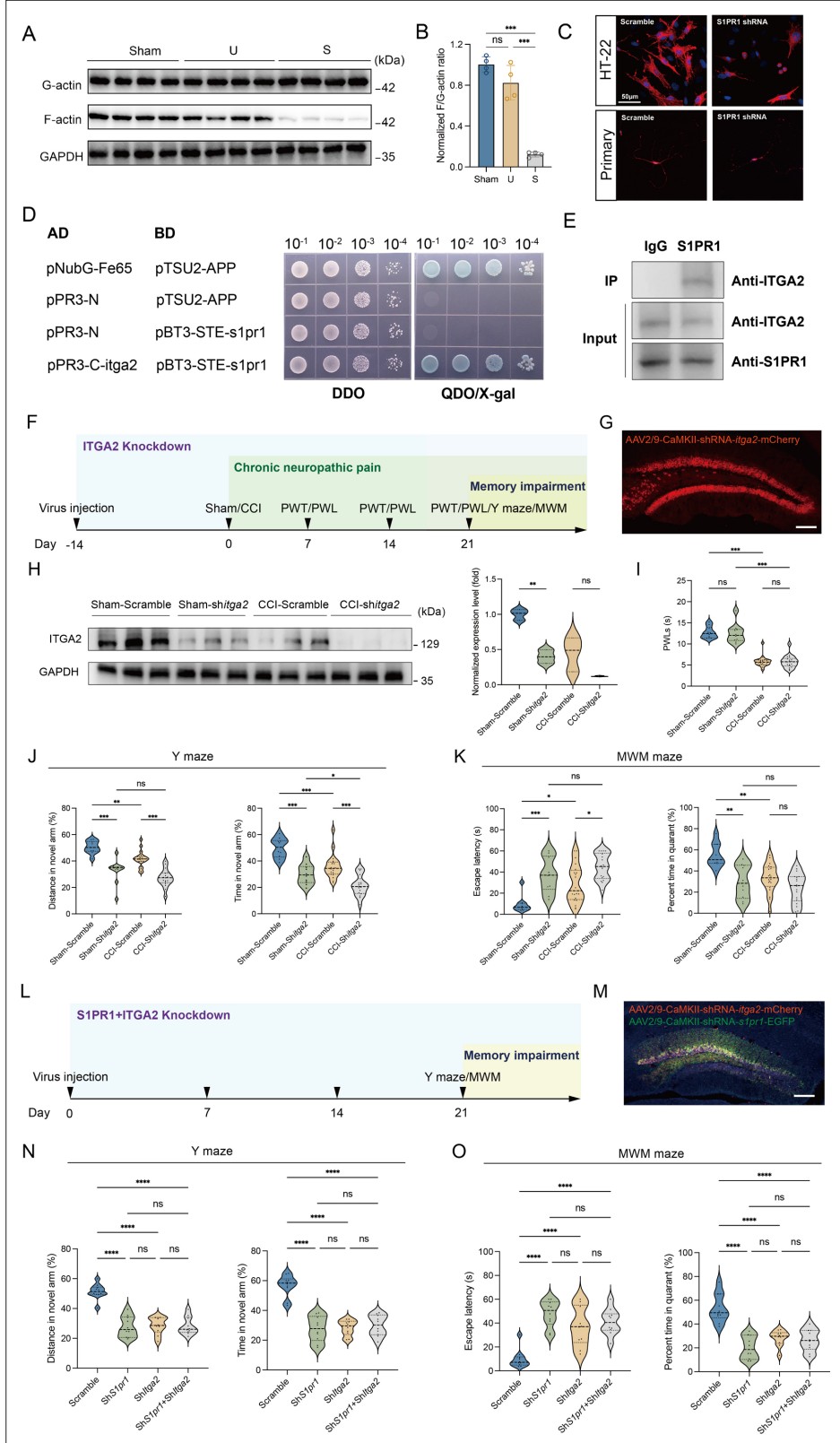

**Figure 9.** S1PR1 regulates actin polymerization by interaction with ITGA2. (**A, B**) Quantification of F/G-actin ratio in dentate gyrus of Sham, U, and S mice by western blot. (**C**) Phalloidin staining of F-actin showing the simple cytoskeleton of *S1pr1*−/− knockdown HT-22 cells and primary hippocampal neurons in comparison to highly organized actin fibers present within scramble HT-22 cells and primary hippocampal neurons (scale bar, 50 μm). (**D**)

*Figure 9 continued on next page*

*Figure 9 continued*

Interaction between S1PR1 and ITGA2 in a yeast two-hybrid system. pNubG-Fe65 and pTSU2-APP were used as a pair of positive control. pPR3-N and pTSU2-APP were used as a negative control. DDO, SD/-Trp/-Leu; QDO, SD/-Trp/-Leu/-His/-Ade. (**E**) In vivo co-immunoprecipitation assay shows that S1PR1 interacts with ITGA2 in the dentate gyrus of mice. Total protein extracts were immunoprecipitated by the anti-S1PR1-specific antibody and analyzed by immunoblot probed with the anti-S1PR1 and anti-ITGA2 antibodies. Immunoglobulin G was used as the negative control. (**F**) Timeline of intra-DG virus injection, CCI surgery, pain threshold tests, Y-maze test, and MWM training. (**G**) A confocal image showing virus expression in the DG (scale bar, 100 µm). (**H**) Example western bands showing efficient ITGA2 knockdown in the DG lysates from Sham-Scramble, Sham-sh*Itga2*, CCI-Scramble, and CCI-sh*Itga2*. Densitometric comparison of the average expression of S1PR1 (*n* = 3). (**I**) Pain threshold in Sham- and CCI-treated mice subjected to Scramble/sh*Itga2* in the DG (*n* = 10–20). (**J**) Quantitative summary of Y-maze showing distances traveled and time spent in the novel arm in Sham- and CCI-treated mice subjected to Scramble/sh*Itga2* in the DG (*n* = 10–19). (**K**) Quantitative summary of MWM training showing escape latency and time spent in the quadrant in Sham- and CCI-treated mice subjected to Scramble/sh*Itga2* in the DG (*n* = 9–20). (**L**) Timeline of intra-DG virus injection, Y-maze test and MWM training. (**M**) A confocal image showing virus expression in the DG (scale bar, 100 µm). (**N**) Quantitative summary of Y-maze showing distances traveled and time spent in the novel arm in WT mice subjected to Scramble/sh*S1pr1*/sh*Itga2*/sh*S1pr1*+sh*Itga2* in the DG (*n* = 10). (**O**) Quantitative summary of MWM training showing escape latency and time spent in the quadrant in WT mice subjected to Scramble/sh*S1pr1*/sh*Itga2*/sh*S1pr1*+sh*Itga2* in the DG (*n* = 10). Data were analyzed by one-way or two-way analysis of variance (one-way or two-way ANOVA), followed by post hoc Tukey's multiple comparisons between multiple groups. All data are presented as the mean ± SEM. ns, not significant; *p < 0.05; **p < 0.01; ***p < 0.001; ****p < 0.0001. CCI, chronic constrictive injury; d, day; MWM, Morris water maze; PWL, paw withdrawal latency; PWT, paw withdrawal threshold; DG, dentate gyrus; U, unsusceptible; S, susceptible.

The online version of this article includes the following source data and figure supplement(s) for figure 9:

**Source data 1.** PDF file containing original western blots for ***Figure 9A, E, H***, indicating the relevant bands.

**Source data 2.** Original files for western blot analysis displayed in ***Figure 9A, E, H***.

**Figure supplement 1.** Verification for plasmid construction by polymerase chain reaction (PCR) followed by restriction digestion.

**Figure supplement 1—source data 1.** PDF file containing unedited gels for ***Figure 9—figure supplement 1***, with lanes labelled.

**Figure supplement 1—source data 2.** Original unedited gels for ***Figure 9—figure supplement 1***.

**Figure supplement 2.** The auto-activation test.

**Figure supplement 3.** Zoomed-out images of the brain to show the precision of the virus injection (scale bar, 1000 µm).

**Figure supplement 4.** Zoomed-out images of the brain to show the precision of the virus injection (scale bar, 1000 µm).

---

the role of S1P signaling in higher pain centers is poorly understood (***Squillace et al., 2020***). The present study provides evidence that in the hippocampal DG, S1P/S1PR1 signaling is irrelevant to pain perception.

Then how does S1PR1 regulate the structural plasticity in the DG and further affect memory formation in chronic neuropathic pain? Our study provides a notable insight by uncovering an integrin α2-dependent modification of the actin cytoskeleton through the activation of the Rac1/Cdc42 signaling cascade and Arp2/3-dependent actin polymerization. Actin cytoskeleton rearrangement is associated with synapse formation, spine architecture and function, thus affecting numerous processes such as memory formation (***Korobova and Svitkina, 2010***; ***Lamprecht and LeDoux, 2004***; ***Bailey et al., 2015***). Rho GTPases (RhoA, RAC1, and CDC42) are key regulators of cytoskeleton assembly and are masters in maintaining spine morphology and memory (***Basu and Lamprecht, 2018***). For example, RAC1 induces branching of actin filaments in lamellipodia by mediating actin polymerization via activating the Arp2/3 complex (***Blanchoin et al., 2000***; ***Goley and Welch, 2006***; ***Xu et al., 2020***). CDC42 is also an important signaling protein for reorganization of actin cytoskeleton and morphogenesis of cells. Loss of CDC42 causes deficits in synaptic plasticity and remote memory recall (***Kim et al., 2014***). Given the evidence that S1P/S1PR1 modulates significant cytoskeletal rearrangements via actin regulatory proteins of Rho GTPases RAC1 and CDC42, but not through RhoA which interacts with S1PR2 (***Cui et al., 2022***; ***Donati and Bruni, 2006***; ***Reinhard, 2017***), we therefore examined the

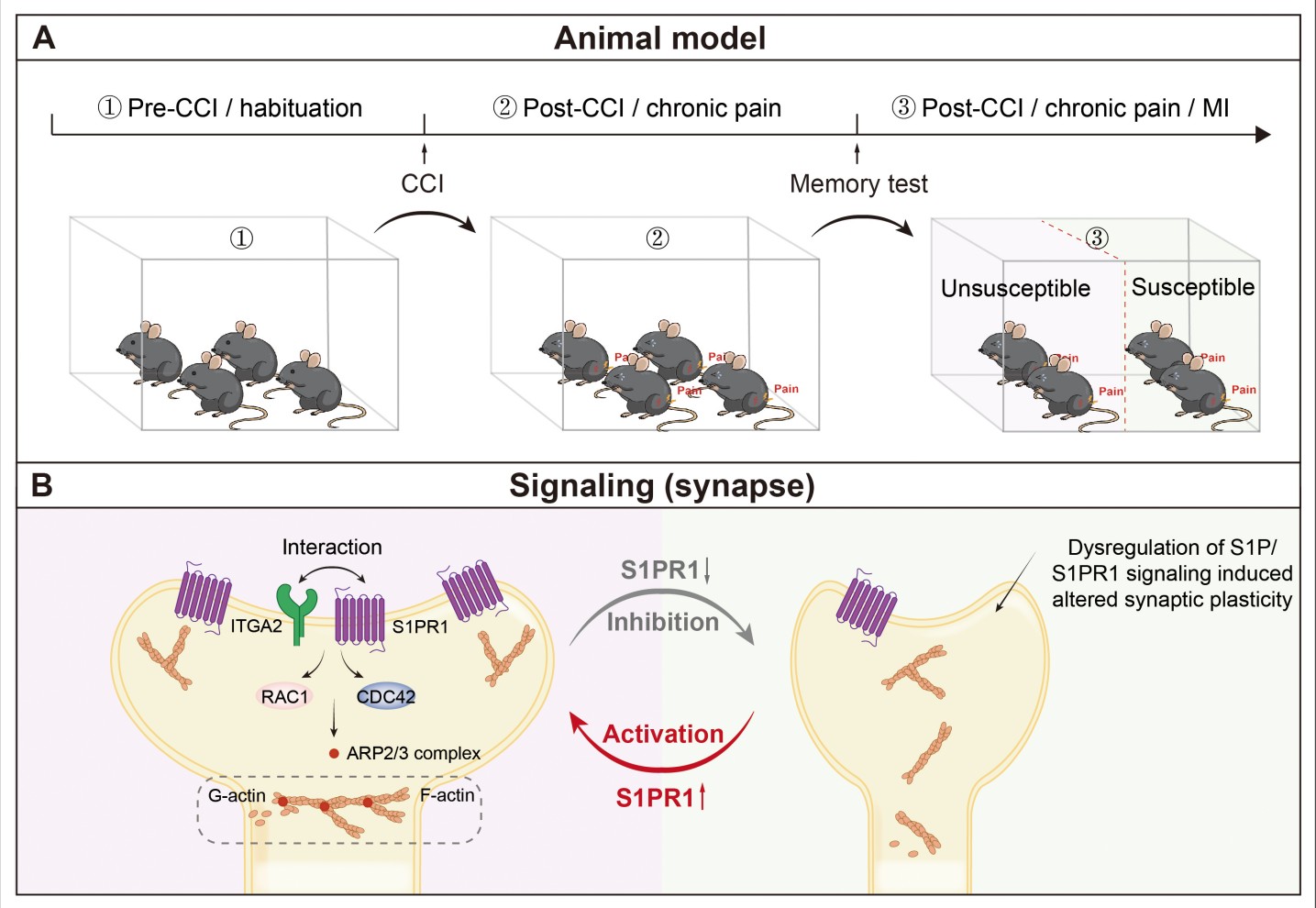

**Figure 10.** Schematic representation of S1P/S1PR1-mediated susceptibility to chronic pain-related memory impairment. (**A**) Mice subjected to chronic pain can be separated into memory impairment-susceptible and -unsusceptible subpopulations 21 days post CCI. (**B**) Structural synaptic plasticity is mainly regulated by the actin cytoskeleton organization. In the DGCs of Sham/unsusceptible mice, S1P/S1PR1 signaling mediates actin dynamics via Itga2-dependent activation of the Rac1/Cdc42 signaling cascade and Arp2/3-dependent actin polymerization, whereas in susceptible mice dysregulation of S1P/S1PR1 signaling in the DGCs leads to defective actin cytoskeleton organization which alters the synaptic plasticity. CCI, chronic constrictive injury; MI, memory impairment; DGCs, dentate granule cells.

expression of RAC1 and CDC42 at protein levels. Our findings revealed a significant downregulation of Rac1 and Cdc42 expression in susceptible mice and mice with S1PR1 knockdown in the DG, suggesting that S1PR1 functions upstream of Rac1 and Cdc42 in the DG. Consistent with this observation, we detected a downregulation of the actin polymerization modulator Arp2/3 complex. Actin remodeling involves the dynamic alterations in actin polymerization that contribute to the structural changes observed at neural synapses. Actin polymerization of the transition of G- into F-actin is crucial for orchestrating all the modifications necessary to facilitate synaptic communication by increasing spine volume. Conversely, the inability to form F-actin results in contrasting outcomes of decreased spine volume and diminished synaptic communication (*Dillon and Goda, 2005*). To further elucidate the causal relationship between DG S1PR1 and actin remodeling, we quantitated F- and G-actin in the mouse DG, and generated a S1PR1 knockdown HT-22 cell line to visualize F-actin by phalloidin staining. Our findings in susceptible mice indicate a significant inhibition of the transition from G- to F-actin, suggesting the involvement of S1P/S1PR1 in DG actin polymerization. RNA-Seq data further revealed that integrin α2 may participate in this regulatory process. Integrins are linkers of the extracellular matrix and intracellular actin cytoskeleton mediating cytoskeletal organization (*Miranti and Brugge, 2002*). The concomitant alteration in expression levels of Itga2 with S1PR1, and the physical

and functional analysis between the two proteins of S1PR1 and ITGA2, indicate their collaboration in the regulation of cytoskeleton arrangement.

This study has potential limitations. First, the memory performance in the chronic pain model is exclusively evaluated based on spatial cues which mainly rely on the hippocampal functions. Further validation using various types of memory tests would strengthen the evidence for the categorization of mice into susceptible and unsusceptible subgroups. Additionally, the study would also benefit from a comprehensive exploration on the impact of different types of chronic pain on memory impairment. Different chronic pains including chronic nociceptive pain, chronic neuropathic pain, composite pain, and chronic psychological pain have different pathological mechanisms and impacts on memories. It is worth noting that preclinical and clinical research has demonstrated that the influence of pain intensity on memory function in chronic pain patients remains controversial (*Liu et al., 2014*). Thus, future research needs to facilitate the investigation of differences between the various subtypes of chronic pains and comorbid memory impairment. Furthermore, the reliance on animal models may limit the generalizability of the findings to humans. Based on the current data, decreased serum S1P level could be a potential biomarker for evaluating the susceptibility of memory impairment. In future studies, it would be interesting to examine whether S1P levels in the serum of patients with pain is associated with memory impairment. Besides, except for S1PR1 and ITGA2 there are also other potential molecular targets implicated in this study, more work is needed to define their roles on the disease occurrence.

In summary, our study develops a paradigm for separating the susceptible and unsusceptible subgroups to chronic pain-induced memory impairment and delineates the key role of S1P/S1PR1 signaling in vulnerability to memory impairment. Since there are currently few pharmacological alternatives for the treatment of the comorbidity, these findings may serve as a foundation for the development of optimal preventive and therapeutic medications. Additionally, the strategy this study used to categorize memory impairment susceptibility may inspire new methods for stratifying patient populations in clinical settings.

## Methods

### Animals

Male C57BL/6J mice (8–10 weeks old) were housed 4–6 per cage with ad libitum access to food and water, at a constant temperature (22– 25°C) and humidity (40–60%). They were kept under a 12:12 light/dark cycle (lights on at 8:00 AM). All animal protocols were approved by the Animal Care and Use Committee of Xuzhou Medical University (202209S054, 202309T017).

### Virus vectors, chemicals, and antibodies

Information for viral vectors, chemicals, and antibodies used in the present study is listed in *Supplementary file 1*.

### Chronic neuropathic pain model

Chronic constrictive injury (CCI) of the sciatic nerve was used to establish chronic neuropathic pain, as described previously (*Bennett and Xie, 1988*). Briefly, mice were anesthetized by 1% pentobarbital sodium (40 mg/kg, *i.p.*) and the left sciatic nerve was exposed by separating the muscles at the mid-thigh level then loosely ligated with three non-absorbable 4-0 slik braided suture proximal to the trifurcation at a 1.0- to 1.5-mm interval. After suturing and sterilizing, animals were placed in a recovery cage on a heating pad. For the following behavioral tests, a control group and an experimental group are identical in all respects of handling and experimental conditions, except for one difference-nerve injury.

### Paw withdrawal threshold

Mechanical allodynia was measured by paw withdrawal threshold PWT in a double-blinded manner by using the up and down method with von Frey filaments. Mice were placed on a wire mesh grid and allowed for 60-min acclimation. A von Frey hair weighing 0.008–2.0 g was used. The filaments (from 0.16 g filament) were perpendicularly applied to the plantar surface of the left hind paw through the wire mesh grid. Once a positive response is observed, change the filament to next lowest level. In the

absence of a response, move to the next highest filament. Paw withdrawal, flinching, or paw licking was considered a positive response. 50% PWT was determined using the 'up-down' method.

## Paw withdrawal latency

Thermal hyperalgesia was evaluated by paw withdrawal latency (PWL) in a double-blinded manner following *Hargreaves et al., 1988*. Briefly, the mice were placed on a glass platform and allowed for 60-min acclimation. Thermal stimulation was focused on the plantar surface of the left hind paw through the glass plate. The time taken to withdraw from the heat stimulus is recorded for three times at an interval of 5 min for rest. The average time was considered as the PWL. Thermal stimulation was no more than 20 s.

## Behavioral test battery to assess learning and memory

We followed the golden rule that we start with the least stressful test (Y-maze) and leave the most stressful of all for last (MWM). Additionally, we also ensured that between tests the animals have enough resting time to decrease carryover effects from prior tests. A common order of behavioral tests associated with learning and memory are Y-maze forced alternation (Y-maze in this study), novel object recognition, MWM, radial arm water maze, and Y-maze spontaneous alternation (*Wolf et al., 2016*).

### Y-maze

Testing occurs in a Y-shaped apparatus consisting of three white, enclosed arms with an angle of 120° from each other, each arm measuring 30 cm in length, 6 cm in width, and 15 cm in height, named the arms of the maze A, B, and C. The experiment was started after the mice had habituated for half an hour in the behavioral chamber, and within 10 min of training, after the C-arm (novel arm) was occluded with a white plastic plate, mice were gently placed in the A-arm so that they were free to explore in both A and B arms. The mice were then returned to their home cage and allowed to move freely for 1 hr. During the test period (5 min), the C-arm (novel arm) was opened, and the time and distance of the mice's exploration in the three arms were traced by a video-tracking system. The percent time and distance of mice exploring in the C-arm (novel arm) were analyzed.

### MWM test

MWM test was performed in a round tank (100 cm in diameter and 51 cm in depth) with a small platform (8.5 cm in diameter, submerged 0.5 cm below water surface). The water was dyed with nontoxic and edible titanium dioxide. A computerized video-tracking system (ANY-maze, Stoelting Co., IL, USA) was located 200 cm above the center of the tank to record performance. The tank was divided into four equal quadrants, southwest (SW), northwest (NW), northeast (NE) and southeast (SE). The platform is placed in the center of SW quadrant. Different graphic cues were put up on the walls of the four orientations to help the mice to build up spatial memory. Mice were trained for 4 days and tested for 2 days. During acquisition training, four trials were conducted every day, starting from the NE quadrant. Animals were placed in the water with its head facing the wall of the tank. The time (s) when the animal touched the underwater platform was recorded. If the time exceeds 60 s, guide the animal to the platform and keep them stay for 10 s. Hereafter, the animals were dried and placed into a care cage. The interval between the two training sessions is 15–20 min. On the first test day, escape latency assay and probe trials were given. The animals were placed in the diagonal quadrant (NE) of the platform quadrant, and the time to reach the platform and the swimming path of the mice were recorded. On the second text day, the platform was removed, and the percent time of total 30 s of each animal spent in the target quadrant and the number of platform-site crossovers were recorded as the index of spatial memory.

## Transmission electron microscope

Mice were deeply anesthetized by 1% pentobarbital sodium (40 mg/kg, *i.p.*) and their brains were collected on the ice to extract DG of hippocampus(1 mm × 1 mm × 0.5 mm) within 1 min. After primary fixation with 4% paraformaldehyde and 3% glutaric dialdehyde solution at 4°C overnight and post-fixation with 1% osmium tetroxide in 0.1 M sodium phosphate buffer at 37°C for 2 hr, tissues were washed with ddH$_2$O four times for 10 min each time. Different concentrations of ethanol and acetone

are used to dehydrate the tissues gradiently. Then they were embedded with paraffin at different temperatures for 48 hr. Semithin sections at a 1-μm thickness were prepared and stained with toluidine blue as survey sections and are viewed using a light microscope in order to locate and trim the regions of hippocampal granule cell layer. Subsequently, ultrathin sections at a 70 nm were obtained and sequentially stained with uranyl acetate and lead citrate for TEM observation. Four to six images for each animal were recorded at ×6300 magnification, and a counting frame (5 × 5 μm$^2$) was placed on each image for counting the number of excitatory synapses and measuring PSD width and length.

## Golgi staining

Golgi staining was performed using FD Rapid Golgi Stain (FD NeuroTechnologies). Mice were anesthetized by 1% pentobarbital sodium (40 mg/kg, *i.p.*). Dissected mouse brains were immersed in solutions A and B in brown bottles for 14 days. The bottles were rotated occasionally to ensure perfect immersion. Fourteen days later, brains were transferred to solution C for 48 hr at 4°C in the dark. The brains were sliced using a vibratome (Leica) at a thickness of 100 μm and mounted on gelatin coated slides. After reimmersion in solution C for 2 min and subsequent air drying, the brain slices were washed twice for 4 min each with ddH$_2$O and stained with solutions D and E for 10 min. Stained brain slices were washed again with ddH$_2$O twice for 4 min each and then gone through gradient dehydration with gradient alcohol (50%, 75%, 95%, 100% ethanol, each 4 min). Fully dehydrated brain slices were washed with xylene three times for 4 min each and then mounted by coverslips using neutral gum as mounting media. Slides were stored in room temperature in the dark. Images of dendritic spines were captured using open field fluorescence microscope and analyzed with ImageJ and Sholl (*Zhang et al., 2023*).

## Immunohistochemistry

Mice were deeply anesthetized by 1% pentobarbital sodium (40 mg/kg, *i.p.*) and sequentially perfused with phosphate-buffered saline (PBS) followed by 4% paraformaldehyde solution. The brains were removed, postfixed with 4% paraformaldehyde solution at 4°C overnight and dehydrated in 30% sucrose solution for 48 hr. Brain slices were prepared coronally with freezing microtome (VT1000S, Leica Microsystems) at a 30-μm thickness and were incubated with the blocking buffer containing 1% bovine serum albumin and 0.1% Triton X-100 in PBS for 2 hr at room temperature and then overnight at 4°C with the primary antibodies. After the slices were washed with Tris-buffered saline (TBS) three times for 10 min each time, they were incubated with secondary antibodies for 2 hr at room temperature and followed by washing with PBS and mounting. After staining, slices were visualized with the laser scanning confocal microscopy (LSM880; Zeiss).

## Western blotting

Hippocampal DG was rapidly stripped and sonicated (Bioruptor UCD-200) in RIPA buffer (P0013B, Beyotime Biotechnology) with 1% cocktail (B14001, Bimake) followed by centrifuge (12,000 rpm, 15 min). Two DGs from one animal were used for each sample. Proteins were then electrophoresed in a 10% sodium dodecyl sulfate–polyacrylamide gel (PG112, Shanghai EpiZyme Scientific) and transferred onto a Polyvinylidene fluoride (PVDF) membrane (IPVH00010, Millipore). The membranes were rinsed in triple and blocked with 5% skim milk before incubation with primary antibodies (*Supplementary file 1c*) at 4°C for 12 hr. After washing with Tris Buffered Saline with Tween 20 (TBST), membranes were incubated with secondary antibodies (*Supplementary file 1c*) for 2 hr at room temperature. The final visualization of the membranes was achieved by ECL Chemiluminescent (BLH01S100CN, Bioworld). We then used housekeeping protein normalization for normalizing western blot data. GAPDH was used as the internal control. The stained blot is imaged, a rectangle is drawn around the target protein in each lane, and the signal intensity inside the rectangle is measured by using ImageJ. The signal intensity obtained can then be normalized by being divided by the signal intensity of the loading internal control (GAPDH) detected on the same blot. The average of the ratios from the control group is calculated, and all individual ratios are divided by this average to obtain a new set of values, which represent the normalized values.

## RNA-Seq

Sham, unsusceptible, and susceptible mice were sacrificed and the DG was rapidly dissected under RNase-free conditions. Total RNA was extracted using Trizol reagent kit (Invitrogen, Carlsbad, CA,

USA) according to the manufacturer's protocol. RNA quality was assessed on an Agilent 2100 Bioanalyzer (Agilent Technologies, Palo Alto, CA, USA) and checked using RNase free agarose gel electrophoresis. After total RNA was extracted, eukaryotic mRNA was enriched by Oligo(dT) beads. Then the enriched mRNA was fragmented into short fragments using fragmentation buffer and reversely transcribed into cDNA by using NEBNext Ultra RNA Library Prep Kit for Illumina (NEB #7530, New England Biolabs, Ipswich, MA, USA). The purified double-stranded cDNA fragments were end repaired, A base added, and ligated to Illumina sequencing adapters. The ligation reaction was purified with the AMPure XP Beads (1.0X), and polymerase chain reaction (PCR) amplified. The resulting cDNA library was sequenced using Illumina Novaseq6000 by Gene Denovo Biotechnology Co. (Guangzhou, China).

## Stereotaxic surgeries

Mice were deeply anesthetized by 1% pentobarbital sodium (40 mg/kg, *i.p.*) and mounted on a stereotaxic apparatus (RWD). A midline incision was made on the scalp after disinfection. The skull was leveled and drilled a small hole in the skull with a dental drill. Virus was bilaterally injected into the DG of hippocampus (−1. 95A/P, ±1. 3M/L, and −2.02 D/V) by a 5-μl Hamilton syringe at a rate of 0.1 μl/min by a microinjection pump (Harvard Apparatus, Holliston, MA). For the infusion of S1PR1 agonist, guide cannula (internal diameter 0.34 mm, RWD) was unilaterally implanted into DG of hippocampus (−1. 95A/P, +1. 3M/L, and −2.02 D/V). After surgery, the mice were placed on a hot blanket for 1 hr and returned to the home cages.

## Local infusion

To deliver S1PR1 agonist SEW2871 (0.7 μM in 200 nl) to the hippocampal DG, 10 μl syringe (Hamilton) connected to an internal stainless-steel syringe and infusion pump inserted into a guide cannula were used. SEW2871 was delivered into the right DG of hippocampus at a flow rate of 100 nl/min. For multiple infusions, mice were deeply anesthetized by 1% pentobarbital sodium (40 mg/kg, *i.p.*).

## Cell culture, transfection, and flow cytometry sorting

HT-22 cells (HT-22 mouse hippocampal neuronal cell line, a gift from Prof. Wuyang Wang, Xuzhou Medical University, the identity has been authenticated by short tandem repeat (STR) profiling and mycoplasma contamination is excluded using PCR) were maintained in Dulbelcco's modified Eagle's medium supplemented with 10% (vol/vol) fetal bovine serum (Invitrogen) in 5% $CO_2$ atmosphere at 37°C. The cell culture was around 70% confluent before transfection in a 6-well plate. Cells were incubated with transfection solution containing viruses (AAV-U6-shRNA(scramble)-GFP and AAV-U6-shRNA(*S1pr1*)-GFP) and polyrene in serum-free medium at 37°C for 8 hr. The media were changed to fresh growth medium with serum post transfection. The transfection efficiency was determined as the percentage of cells expressing GFP in the entire cell population. In order to harvest the stable S1pr1 knock-down virus transfected cell line, we conducted flow cytometry sorting. Briefly, we discarded the original medium of the adherent cultured AAV-U6-shRNA(scramble)-GFP- and AAV-U6-shRNA(*S1pr1*)-GFP-transfected HT-22 cells, washed with sterilized 1× PBS once, added ethylenediaminetetraacetic acid (EDTA)-free trypsin to digest for 3 min, and terminated the digestion by an equal volume of serum-containing medium. Subsequently, the cell suspension was collected, centrifuged at 600 rpm for 4 min. The supernatant was discarded and resuspended with sterilized 1xPBS into flow tubes. GFP-positive cells (transfected cells) were sorted out by flow cytometry (Beckman CytoFLEX SRT, Beckman) for subsequent experimental processes.

## In vivo co-immunoprecipitation assay

DG tissues were carefully dissected, lysed in RIPA buffer (P0013B, Beyotime Biotechnology) with 1× Protease inhibitor cocktail (B14001, Bimake), sonicated (Diagenode, UCD-200) and centrifuged at 12,000 rpm for 15 min at 4°C. The supernatant was taken for the co-immunoprecipitation assay using BeaverBeads Protein A Immunoprecipitation Kit (Beaver, 22202-20) following the manufacturer's instructions. Briefly, 30 μl Protein A beads were mixed with 5 μg/ml anti-S1PR1 antibody (Abcam, Ab259902) and incubated with gentle rotation at room temperature for 15 min. Beads were pelleted and washed twice. Then the beads and the supernatant were incubated with rotation at room temperature for 1 hr. The bound proteins were eluted from the beads with 1× Protein Loading

Buffer by boiling at 95°C for 5 min. Protein samples were then separated and immunoblotted with anti-ITGA2 antibody (Bioss, bsm-52613R) by western blot.

## Split-ubiquitin membrane-based yeast two-hybrid system

In this ubiquitin system, ubiquitin, a protein composed of 76 amino acid residues that can mediate the ubiquitination degradation of target proteins by proteasomes, is split into two domains, namely Cub at the C-terminus and NbuG at the N-terminus, which are fused and expressed with the bait protein 'Bait' and the prey protein 'Prey', respectively. Here, the coding region of s1pr1 was fused to the 'bait' pBT3-STE vector (Clontech) cloning at Sfi IA (5'-GGCCATTACGGCC-3') and Sfi IB (3'-GGCC GCCTCGGCC-5') sites. Full length of itga2 was inserted into the Sfi IA (5'-GGCCATTACGGCC-3') and Sfi IB (3'-GGCCGCCTCGGCC-5') sites of the 'prey' pPR3-C vector (Clontech). At the same time, Cub is also fused with transcription factor LEXa-VP16. If Bait and Prey proteins could bind, Cub and NbuG would be brought together and a complete ubiquitin would be formed, which would be recognized by the proteasome and the fused transcription factor would be cut off and enter the cell nucleus to activate the expression of the reporter gene. Series of combinations of bait and prey constructs were cotransformed into the yeast strain NMY51 (Clontech), and different concentrations of bacterial liquid were growing onto SD/-Trp/-Leu plates for 3 days at 30°C, interactions between baits and preys were examined on the selective medium SD/-Leu/-Trp/-His/-Ade. The blue colonies were chosen as candidates for possible interaction. The pNubG-Fe65 and pTSU2-APP vectors were served as positive controls, while the pPR3-N and pTSU2-APP were served as a negative control. Detailed procedures were conducted following the manufacturer's instructions (Clontech).

## k-Means algorithm

k-Means algorithm is used to partition the dataset into pre-defined number of clusters. Each data point belongs to a single group based on the distance between their centroids. The centroid is either the mean or median of all the points within the cluster depending on the characteristics of the data. The main procedures for k-means algorithm are as follows: (1) determine the k, namely group numbers; (2) randomly assign a centroid to each of the k clusters; (3) calculate the distance of all data to each of the k centroids; (4) assign data to the closest centroid; (5) update the centroid by taking the mean of all the points in each cluster; (6) repeat steps (3)–(5) until convergence; (7) the algorithm outputs the final cluster centroids and the assignment of each data point to a cluster. For categorization the mice into susceptible and unsusceptible groups (two groups), we analyzed results from Y-maze and MWM tests using the k-means clustering algorithm and determine the k as 2. After assigning each data point to its closest k-center, we drew a median between both the centroids as a cutoff value. One cluster, including mice displaying a ratio of time more than the cutoff value, was defined as the unsusceptible mouse cluster. The other cluster, including mice exhibiting a ratio of time less than the cutoff value, was defined as the susceptible mouse cluster.

## Statistics

The mice were assigned randomly to either the control group or the experimental group for each experiment. Investigators responsible for the behavioral test are blinded to which animals represent treatments or controls. Data from mice with signs of illness or locomotion dysfunction will be excluded from our analysis. However, none of these issues were observed in our study. Data are presented as mean ± SEM. The summarized data in the violin plots were presented as the median (indicated by the bold dash line) along with the 25th and 75th percentiles (indicated by the slim dash line). k-Means cluster analysis was used to partition a given dataset into a set of k groups (in this study, k = 2). Unpaired Student's t test was used to compare the mean of two independent groups. One-way analysis of variance (ANOVA) with post hoc Tukey's multiple comparisons was used to examine the differences in the means of three or more groups. Two-way ANOVA with post hoc Tukey's multiple comparisons was employed to determine the mean of a quantitative variable changes according to the levels of two independent variables. Statistical significance is indicated by asterisks as *p < 0.05, **p < 0.01, and ***p < 0.001. Statistical analyses were performed using GraphPad Prism 9.0. and SPSS V22. Notably, every experiment was carried out a minimum of three times to make sure the results were consistent throughout the manuscript.

# Additional information

## Funding

| Funder | Grant reference number | Author |
|---|---|---|
| National Sci-Tech Innovation-2030 Major Projects | 2021ZD0203100 | Jun-Li Cao |
| National Natural Science Foundation of China | 82130033 | Jun-Li Cao |
| National Natural Science Foundation of China | 82293641 | Jun-Li Cao |
| National Natural Science Foundation of China | 31970937 | Hongxing Zhang |
| National Natural Science Foundation of China | 82271255 | Hongxing Zhang |
| National Natural Science Foundation of China | 82101315 | Mengqiao Cui |
| National Natural Science Foundation of China | 82204081 | Weiyi Song |
| Natural Science Foundation of Jiangsu Province | BK20220665 | Weiyi Song |
| Natural Science Foundation of Jiangsu Province | BK20210908 | Zhou Wu |
| Priority Academic Program Development of Jiangsu Higher Education Institutions | 21KJB320001 | Mengqiao Cui |
| China Postdoctoral Science Foundation | 2022M722676 | Mengqiao Cui |
| China Postdoctoral Science Foundation | 2022M722675 | Weiyi Song |
| Xuzhou Medical University | D2020053 | Mengqiao Cui |
| Xuzhou Medical University | D2020033 | Zhou Wu |
| Xuzhou Medical University | TD202203 | Mengqiao Cui |
| Fusion Innovation Foundation of Xuzhou Medical University | XYRHCX2021009 | Weiyi Song |
| Jiangsu Basic Research Programs | BK20243035 | Jun-Li Cao |
| Key Technologies R&D Program of Guangdong Province | 2023B0303020003 | Jun-Li Cao |
| National Natural Science Foundation of China | 82271263 | Hai-Lei Ding |
| Key Project of Nature Science Foundation of Jiangsu Education Department | 22KJA320006 | Hai-Lei Ding |

The funders had no role in study design, data collection, and interpretation, or the decision to submit the work for publication.

## Author contributions
Mengqiao Cui, Conceptualization, Data curation, Software, Formal analysis, Funding acquisition, Investigation, Methodology, Writing - original draft, Project administration; Xiaoyuan Pan, Zhijie Fan, Data curation, Software, Formal analysis, Investigation, Methodology, Writing - original draft; Shulin Wu, Data curation, Formal analysis, Investigation, Methodology; Ran Ji, Xianlei Wang, Xiangxi Kong, Jun-Xia Yang, Data curation, Investigation; Zhou Wu, Lingzhen Song, Weiyi Song, Data curation, Funding acquisition, Investigation; Hongjie Zhang, Writing – review and editing; Hongxing Zhang, Conceptualization, Supervision, Funding acquisition, Writing – review and editing; Hai-Lei Ding, Conceptualization, Supervision, Project administration, Writing – review and editing; Jun-Li Cao, Conceptualization, Supervision, Funding acquisition, Project administration, Writing – review and editing

## Author ORCIDs
Mengqiao Cui (ID) https://orcid.org/0000-0002-7098-8718
Zhou Wu (ID) https://orcid.org/0000-0002-1086-9290
Jun-Li Cao (ID) https://orcid.org/0000-0002-8932-4743

Reviewer #1 (Public review): https://doi.org/10.7554/eLife.99862.3.sa1
Reviewer #2 (Public review): https://doi.org/10.7554/eLife.99862.3.sa2
Author response https://doi.org/10.7554/eLife.99862.3.sa3

---

# Additional files

## Supplementary files
- Supplementary file 1. Virus vectors, chemicals, and antibodies used in this study.
- MDAR checklist

## Data availability
Sequencing data have been deposited in Sequence Read Archive (SRA) under accession code PRJNA1189590.

The following dataset was generated:

| Author(s) | Year | Dataset title | Dataset URL | Database and Identifier |
|---|---|---|---|---|
| Cui M | 2024 | RNA-seq for dentate gyrus in mice with chronic pain related memory impairment | https://www.ncbi.nlm.nih.gov/sra/PRJNA1189590 | NCBI Sequence Read Archive, PRJNA1189590 |

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
