## [Editor Report · eLife Assessment]

This study investigates the molecular mechanisms underlying chronic pain-related memory impairment by focusing on S1P/S1PR1 signaling in the dentate gyrus (DG) of the hippocampus. Through behavioral tests (Y-maze and Morris water maze) and RNA-seq analysis, the researchers discovered that S1P/S1PR1 signaling is crucial for determining susceptibility to memory impairment, with decreased S1PR1 expression linked to structural plasticity changes and memory deficits. This work has **important** significance and a **convincing** level of evidence, thus offering new insights into the mechanisms underlying chronic pain-related memory impairment.

---

## [Referee Report · Reviewer #1 (Public review)]

This work from Cui, Pan, Fan et al explores memory impairment in chronic pain mouse models, a topic of great interest for the neurobiology field. In particular, the work starts from a very interesting observation, that WT mice can be divided in susceptible and unsusceptible to memory impairment upon modelling chronic pain with CCI. This observation represents the basis of the work where the authors identify the sphingosine receptor S1PR1 as down-regulated in the dentate gyrus of susceptible animals and demonstrate through an elegant range of experiments involving AAV mediated knockdown or overexpression of S1PR1 that this receptor is involved in the memory impairment observed with chronic pain. Importantly for translational purposes, they also show that activation of S1PR1 through a pharmacological paradigm is able to rescue the memory impairment phenotype.

The authors also link these defects to reduced dendritic branching and reduced number of mature excitatory synapses in the DG to the memory phenotype.

They then proceed to explore possible mechanisms downstream of S1PR1 that could explain this reduction in dendritic spines. They identify integrin α2 as an interactor of S1PR1 and show a reduction in several proteins involved in actin dynamic, which is crucial for dendritic spine formation and plasticity.

They thus hypothesize that the interaction between S1PR1 and Integrin α2 is fundamental for the activation of Rac1 and Cdc42 and consequently for the polymerisation of actin; a reduction in this pathway upon chronic pain would thus lead to impaired actin polymerisation, synapse formation and thus impaired memory.

The work is of great interest and the experiments are of very good quality with results of great importance.

Comments on revisions:

The authors have replied satisfactorily to my previous concerns.

---

## [Referee Report · Reviewer #2 (Public review)]

Summary:

The study investigates the molecular mechanisms underlying chronic pain-related memory impairment by focusing on S1P/S1PR1 signaling in the dentate gyrus (DG) of the hippocampus. Through behavioural tests (Y-maze and Morris water maze) and RNA-seq analysis, the researchers segregated chronic pain mice into memory impairment-susceptible and -unsusceptible subpopulations. They discovered that S1P/S1PR1 signaling is crucial for determining susceptibility to memory impairment, with decreased S1PR1 expression linked to structural plasticity changes and memory deficits.

Knockdown of S1PR1 in the DG induced a susceptible phenotype, while overexpression or pharmacological activation of S1PR1 promoted resistance to memory impairment and restored normal synaptic structure. The study identifies actin cytoskeleton-related pathways, including ITGA2 and its downstream Rac1/Cdc42 signaling, as key mediators of S1PR1's effects, offering new insights and potential therapeutic targets for chronic pain-related cognitive dysfunction.

This manuscript consists of a comprehensive investigation and significant findings. The study provides novel insights into the molecular mechanisms of chronic pain-related memory impairment, highlighting the critical role of S1P/S1PR1 signaling in the hippocampal dentate gyrus. The clear identification of S1P/S1PR1 as a potential therapeutic target offers promising avenues for future research and treatment strategies. The manuscript is well-structured, methodologically sound, and presents valuable contributions to the field.

Strengths:

(1) The manuscript is well-structured and written in clear, concise language. The flow of information is logical and easy to follow.

(2) The segregation of mice into memory impairment-susceptible and -unsusceptible subpopulations is innovative and well-justified. The statistical analyses are robust and appropriate for the data.

(3) The detailed examination of S1PR1 expression and its impact on synaptic plasticity and actin cytoskeleton reorganization is impressive. The findings are significant and contribute to the understanding of chronic pain-related memory impairment.

Comments on revisions:

The authors have satisfactorily addressed all the issues raised.

---

## [Author Response]

The following is the authors’ response to the original reviews.

**Public Reviews:**

**Reviewer #1 (Public Review):**
This work from Cui, Pan, Fan, et al explores memory impairment in chronic pain mouse models, a topic of great interest in the neurobiology field. In particular, the work starts from a very interesting observation, that WT mice can be divided into susceptible and unsusceptible to memory impairment upon modelling chronic pain with CCI. This observation represents the basis of the work where the authors identify the sphingosine receptor S1PR1 as down-regulated in the dentate gyrus of susceptible animals and demonstrate through an elegant range of experiments involving AAV-mediated knockdown or overexpression of S1PR1 that this receptor is involved in the memory impairment observed with chronic pain. Importantly for translational purposes, they also show that activation of S1PR1 through a pharmacological paradigm is able to rescue the memory impairment phenotype.The authors also link these defects to reduced dendritic branching and a reduced number of mature excitatory synapses in the DG to the memory phenotype.They then proceed to explore possible mechanisms downstream of S1PR1 that could explain this reduction in dendritic spines. They identify integrin α2 as an interactor of S1PR1 and show a reduction in several proteins involved in actin dynamic, which is crucial for dendritic spine formation and plasticity.They thus hypothesize that the interaction between S1PR1 and Integrin α2 is fundamental for the activation of Rac1 and Cdc42 and consequently for the polymerisation of actin; a reduction in this pathway upon chronic pain would thus lead to impaired actin polymerisation, synapse formation, and thus impaired memory.The work is of great interest and the experiments are of very good quality with results of great importance. I have however some concerns. The main concern I have relates to the last part of the work, namely Figures 8 and 9, which I feel are not at the same level as the results presented in the previous 7 Figures, which are instead outstanding.In particular:- In Figure 8, given the reduction in all the proteins tested, the authors need to check some additional proteins as controls. One good candidate could be RhoA, considering the authors say it is activated by S1PR2 and not by S1PR1;

Thanks for your suggestion. We tested the expression level of RhoA in mice 7 days and 21 days post CCI as negative controls (Supplemental Figure 9).

- In addition to the previous point, could the authors also show that the number of neurons is not grossly different between susceptible and unsusceptible mice? This could be done by simply staining for NeuN or performing a western blot for a neuronal-specific protein (e.g. Map2 or beta3-tubulin);

As suggested, we performed immunofluorescence using NeuN antibody to detect the number of neurons in susceptible and unsusceptible mice. The number is not significantly different between the two populations (Supplementary Figure 7).

- In Figure 8, the authors should also evaluate the levels of activated RAC1 and activated Cdc42, which are much more important than just basal levels of the proteins to infer an effect on actin dynamics. This is possible through kits that use specific adaptors to pulldown GTP-Rac1 and GTP-Cdc42;

Thanks for your constructive suggestion. An elevated level and hyperactivation of Rac1 protein are both associated with actin dynamics and dendritic development [1]. We agree that showing the levels of activated RAC1 is better to infer its effect on actin dynamics. Here in Figure 8, the purpose of this experiment is to prove the levels of actin organization related proteins are altered according to the expression level of S1PR1, thus drawing a conclusion that the actin organization was disrupted, but not to specifically emphasize that S1PR1 activated these proteins. We apologize for the confusion made but we think the current data is enough to support the conclusion.

Thanks again for your advice. Your understanding is greatly appreciated.

- In Figure 9C, the experiment is performed in an immortalised cell line. I feel this needs to be performed at least in primary hippocampal neurons;

Thanks for your suggestion. As suggested, we performed the experiment in primary hippocampal neurons. Knockdown of *S1pr1* in primary hippocampal neurons induced reduction in the number of branches and filamentous actin. Please refer to the updated Figure 9C.

- In Figure 9D, the authors use a Yeast two-hybrid system to demonstrate the interaction between S1PR1 and Integrin α2. However, as the yeast two-hybrid system is based on the proximity of the GAL4 activating domain and the GAL4 binding domain, which are used to activate the transcription of reporter genes, the system is not often used when probing the interaction between transmembrane proteins. Could the authors use other transmembrane proteins as negative controls?;

Thanks for your question. We apologize for the unclear description in the method part. Traditional yeast two-hybrid system can only detect protein interactions that occur in the nucleus, but cannot detect ones between membrane proteins. Here, we utilized the split-ubiquitin membrane-based Yeast two-hybrid system. Briefly, in the ubiquitin system, ubiquitin, a protein composed of 76 amino acid residues that can mediate the ubiquitination degradation of target proteins by proteasomes, is split into two domains, namely Cub at the C-terminus and NbuG at the N-terminus, which are fused and expressed with the bait protein “Bait” and the prey protein “Prey”, respectively. At the same time, Cub is also fused with transcription factors. If Bait and Prey proteins could bind, Cub and NbuG would be brought together and a complete ubiquitin would be formed, which would be recognized by the proteasome and the fused transcription factor would be cut off and enter the cell nucleus to activate the expression of the reporter gene. We then determine whether the Bait and Prey proteins interact with each other through the growth of the yeast.

Thanks again for pointing this out. We reworded the method in M&M (Line 678-696).

- In Figure 9E, the immunoblot is very unconvincing. The bands in the inputs are very weak for both ITGA2 and S1PR1, the authors do not show the enrichment of S1PR1 upon its immunoprecipitation and the band for ITGA2 in the IP fraction has a weird appearance. Were these experiments performed on DG lysates only? If so, I suggest the authors repeat the experiment using the whole brain (or at least the whole hippocampus) so as to have more starting material. Alternatively, if this doesn't work, or in addition, they could also perform the immunoprecipitation in heterologous cells overexpressing the two proteins;

Thanks for the question and suggestion. We used DG lysates from both the dentate gyrus of a single mouse as the starting material. We updated the result which showed clearer bands (Figure 9E).

- About the point above, even if the results were convincing, the authors can't say that they demonstrate an interaction in vivo. In co-IP experiments, the interaction is much more likely to occur in the lysate during the incubation period rather than being conserved from the in vivo state. These co-IPs demonstrate the ability of proteins to interact, not necessarily that they do it in vivo. If the authors wanted to demonstrate this, they could perform a Proximity ligation assay in primary hippocampal neurons, using antibodies against S1PR1 and ITGA2.

Thanks for your concern. Co-immunoprecipitation (Co-IP) is the gold standard to identify protein-protein interactions [2], and it is one of the most efficient techniques to study these protein-protein interactions in vivo [3]. We repeated the experiment and followed the experimental procedure exactly to avoid the protein interaction due to over-incubation. Over-incubation, particularly at room temperature, may result in non-specific binding and therefore high background, thus we performed Co-IPs at 4°C to preserve protein interactions. We agree that Proximity ligation assay is better suited for studies of endogenously expressed proteins in primary cells [4]. Since we optimized the experiment procedure to avoid non-specific binding and particularly, Co-IP utilized proteins from DG lysates which could validate the specificity of the protein interaction in native tissue, we prefer to keep the Co-IP result in Figure 9E.

Thanks again for your suggestion. We appreciate your understanding on this matter.

- In Figure 9H, could the authors increase the N to see if shItga2 causes further KD in the CCI?

As suggested, we repeated the experiment and increased the N to 6. As shown in the following picture, shItga2 did not cause further KD in the CCI.

- To conclusively demonstrate that S1PR1 and ITGA2 participate in the same pathway, they could show that knocking down the two proteins at the same time does not have additive effects on behavioral tests compared to the knockdown of each one of them in isolation.

Thanks for your suggestion. As suggested, we knocked down the two proteins at the same and did not observe additive effects on behavioral tests compared to the knockdown of each one of them in isolation. Please refer to Figure 9L-O.

Other major concerns:- Supplementary Figure 5: the image showing colocalisation between S1PR1 and CamKII is not very convincing. Is the S1PR1 antibody validated on Knockout or knockdown in immunostaining?;

S1PR1 is a membrane receptor and the S1P1 antibody (PA1-1040, Invitrogen) shows membranous staining with diffuse dot-like signals (Please refer to the image “A” provided by ThermoFisher Scientific). Here, we utilized the antibody to detect the expression of S1PR1 in DG granule cells. We can see the diffuse dot-like signals aggregated in each single granule cell. CaMKII shows intense staining around the border of the granule cell soma (Image “B”) [5]. According to the images shown in Supplementary Figure 5B, we concluded that S1PR1 is expressed in CaMKII+ cells.

Besides, as suggested, we validated the S1PR1 antibody on knockdown in immunostaining (Image “C” and “D”). The expression of S1PR1 is significantly decreased compared with the control.

**Author response image 2. sa3fig2:** 

- It would be interesting to check S1PR2 levels as a control in CCI-chronic animals;

As suggested, we quantified the S1PR2 levels in Sham and CCI animals, and there is no significant difference between groups (Supplementary Figure 9).

- Figure 1: I am a bit concerned about the Ns in these experiments. In the chronic pain experiments, the N for Sham is around 8 whereas is around 20 for CCI animals. Although I understand higher numbers are necessary to see the susceptible and unsusceptible populations, I feel that then the same number of Sham animals should be used;

Thanks for your concern. In the preliminary experiment, we noticed that the ratio of susceptible and unsusceptible populations is around 1:1. After the behavioral tests, we need to further take samples to investigate molecular and cellular changes of each group. Thus, we set sham around 8 and CCI around 20 to ensure that after characterization into susceptible and unsusceptible groups, each group has relatively equal numbers for further investigations.

- Figures 1E and 1G have much higher Ns than the other panels. Why is that? If they have performed this high number of animals why not show them in all panels?;

Thanks for your concern. For Figure 1B, C, D and F, we showed the data for each batch of experiment, while for Figure 1E and 1G, we used data collected from all batches of experiment. To show the data from a single batch, we would like to demonstrate the ratio of susceptible to unsusceptible is relatively stable, but not only based on a big sample size.

- In the experiments where viral injection is performed, the authors should show a zoomed-out image of the brain to show the precision of the injection and how spread the expression of the different viruses was;

As suggested, we showed the zoomed-out image in Supplementary Figure 6. The viruses are mainly expressed in the hippocampal DG.

- The authors should check if there is brain inflammation in CCI chronic animals. This would be interesting to explain if this could be the trigger for the effects seen in neurons. In particular, the authors should check astrocytes and microglia. This is of interest also because the pathways altered in Figure 8A are related to viral infection.- If the previous point shows increased brain inflammation, it would be interesting for the authors to check whether a prolonged anti-inflammatory treatment in CCI animals administered before the insurgence of memory impairment could stop it from happening;- In addition, the authors should speculate on what could be the signal that can induce these molecular changes starting from the site of injury;- Also, as the animals are all WT, the authors should speculate on what could render some animals prone to have memory impairments and others resistant.

Thanks for the above four suggestions. We have observed inflammation including T cell infiltration and microglia activation in the hippocampal DG in CCI chronic animals and also used S1PR1 modulator which has anti-lymphocyte mediated inflammatory effect to prevent the insurgence of memory impairment from happening. We also examined the alteration in the numbers of peripheral T-lymphocyte subsets and the serum levels of cytokines. Furthermore, we found a neuron-microglia dialogue in the DG which may promote the resilience to memory impairment in CCI animals. Since these are unpublished results, we apologize that we would not give much detailed information to the public at the current stage. We will publish these data as soon as possible. Thanks for your understanding.

**Reviewer #2 (Public Review):**
Summary:The study investigates the molecular mechanisms underlying chronic pain-related memory impairment by focusing on S1P/S1PR1 signaling in the dentate gyrus (DG) of the hippocampus. Through behavioural tests (Y-maze and Morris water maze) and RNA-seq analysis, the researchers segregated chronic pain mice into memory impairment-susceptible and -unsusceptible subpopulations. They discovered that S1P/S1PR1 signaling is crucial for determining susceptibility to memory impairment, with decreased S1PR1 expression linked to structural plasticity changes and memory deficits.Knockdown of S1PR1 in the DG induced a susceptible phenotype, while overexpression or pharmacological activation of S1PR1 promoted resistance to memory impairment and restored normal synaptic structure. The study identifies actin cytoskeleton-related pathways, including ITGA2 and its downstream Rac1/Cdc42 signaling, as key mediators of S1PR1's effects, offering new insights and potential therapeutic targets for chronic pain-related cognitive dysfunction.This manuscript consists of a comprehensive investigation and significant findings. The study provides novel insights into the molecular mechanisms of chronic pain-related memory impairment, highlighting the critical role of S1P/S1PR1 signaling in the hippocampal dentate gyrus. The clear identification of S1P/S1PR1 as a potential therapeutic target offers promising avenues for future research and treatment strategies. The manuscript is well-structured, methodologically sound, and presents valuable contributions to the field.Strengths:(1) The manuscript is well-structured and written in clear, concise language. The flow of information is logical and easy to follow.(2) The segregation of mice into memory impairment-susceptible and -unsusceptible subpopulations is innovative and well-justified. The statistical analyses are robust and appropriate for the data.(3) The detailed examination of S1PR1 expression and its impact on synaptic plasticity and actin cytoskeleton reorganization is impressive. The findings are significant and contribute to the understanding of chronic pain-related memory impairment.Weaknesses:(1) Results: While the results are comprehensive, some sections are data-heavy and could be more reader-friendly with summarized key points before diving into detailed data.

Thanks for the suggestion. For the first sentence in each part/paragraph, we used statement that summarises what will be investigating in the following experiments to make it more reader-friendly. They are labeled as blue in the main text.

(2) Discussion: There is a need for a more balanced discussion regarding the limitations of the study. For example, addressing potential biases in the animal model or limitations in the generalizability of the findings to humans would strengthen the discussion. Also, providing specific suggestions for follow-up studies would be beneficial.

As suggested, we discussed more on the limitations of this study and outlined some directions for future research (Line 481-498).

(3) Conclusion: The conclusion, while concise, could better highlight the study's broader impact on the field and potential clinical implications.

Thanks. We reworded the conclusion to better highlight the impacts of this study (Line 501-505).

**Reviewer #3 (Public Review):**
Summary of the Authors' Objectives:The authors aimed to delineate the role of S1P/S1PR1 signaling in the dentate gyrus in the context of memory impairment associated with chronic pain. They sought to understand the molecular mechanisms contributing to the variability in memory impairment susceptibility and to identify potential therapeutic targets.Major Strengths and Weaknesses of the Study:The study is methodologically robust, employing a combination of RNA-seq analysis, viral-mediated gene manipulation, and pharmacological interventions to investigate the S1P/S1PR1 pathway. The use of both knockdown and overexpression approaches to modulate S1PR1 levels provides compelling evidence for its role in memory impairment. The research also benefits from a comprehensive assessment of behavioral changes associated with chronic pain.However, the study has some weaknesses. The categorization of mice into 'susceptible' and 'unsusceptible' groups based on memory performance requires further validation. Additionally, the reliance on a single animal model may limit the generalizability of the findings. The study could also benefit from a more detailed exploration of the impact of different types of pain on memory impairment.Assessment of the Authors' Achievements:The authors successfully identified S1P/S1PR1 signaling as a key factor in chronic pain-related memory impairment and demonstrated its potential as a therapeutic target. The findings are supported by rigorous experimental evidence, including biochemical, histological, and behavioral data. However, the study's impact could be enhanced by further exploration of the molecular pathways downstream of S1PR1 and by assessing the long-term effects of S1PR1 manipulation.Impact on the Field and Utility to the Community:This study is likely to have a significant impact on pain research by providing a novel perspective on the mechanisms underlying memory impairment in chronic pain conditions. The identification of the S1P/S1PR1 pathway as a potential therapeutic target could guide the development of new treatments.Additional Context for Readers:The study's approach to categorizing susceptibility to memory impairment could inspire new methods for stratifying patient populations in clinical settings.Recommendations:(1) A more detailed explanation of the k-means clustering algorithm and its application in categorizing mice should be provided.

As suggested, we explained the k-means clustering algorithm in details (Line 697-711).

(2) The discussion on the potential influence of different pain types or sensitivities on memory impairment should be expanded.

Thanks for your suggestion. We discussed this point in the limitations of this study (Line 484-491).

(3) The protocol for behavioral testing should be clarified and the potential for learning or stress effects should be addressed.

Thanks for your suggestion. We clarified the order of the battery of behavioral tests in this study (Line 537-542). We start with the least stressful test (Y-maze) and leave the most stressful of all for last (Morris Water maze) [6]. Besides, we also conducted behavioral assays to prove that a one-day rest is enough to decrease carryover effects from prior test (Y-maze). We examined the stress related behaviors one day after Y-maze (23d post CCI) using open field test (OFT) and elevated plus maze (EPM). As shown in Author response image 3, the tests did not reflect the mice were under stressful circumstances. Thus, the order in which the tests were performed are appropriate in this study.

**Author response image 3. sa3fig3:** 

(4) Conduct additional behavioral assays for other molecular targets implicated in the study.

We agree that other molecular targets on susceptibility to memory impairment would be interesting to know. Our study was designed to focus specifically on ITGA2 this time and we'd like to keep the focus intact, but we have included your point as a consideration for future study (Lines 496-498). Thank you for the suggestion.

(5) The effective drug thresholds and potential non-specific effects of pharmacological interventions should be discussed in more detail.

As suggested, we emphasized this point of drug SEW2871 in Line 242-245.

**Recommendations for the authors:**

**Reviewer #1 (Recommendations For The Authors):**
Minor concerns:- In Figure 6E the lines of the different groups are not visible. Showing the errors as error bars for each point would probably be better;

We apologize for the mistake of using mean ± SD here instead of mean ± SEM. After changing to mean ± SEM, the lines of Figure 6E, Figure 7E and 7L become much clearer. It looks a little bit messy to show the error bars since there are numerous points, so we prefer to keep the line style.

- Do the authors have any speculation on why the % time in the quadrant is not further affected in the KD Itga2 in CCI animals (Figure 9K)?;

In CCI animals, the level of S1PR1 expression is decreased. ITGA2 may participate in the same pathway with S1PR1. Thus, knocking down ITGA2 in CCI animals will not further affect the animal behaviors. This has been proved by knocking down the two proteins at the same time and no additive effects were observed on behavioral tests compared to the knockdown of each one of them in isolation (Figure 9L-O).

- In the methods, it's unclear if in the multiple infusion, the animals were anaesthetised or kept awake;

We have clarified this point in the method. mice were deeply anesthetized by 1% pentobarbital sodium (40 mg/kg, i.p.). (Line 649-650)

- As the DG is quite small, could the authors clarify if, when performing western blots, they used the two DGs from one animal for each sample or if they pulled together the DGs of several animals?;

We used the two DGs from one animal for each sample. The amount of protein extracted from each sample is enough for 20-30 times of Western Blot assays. We have now added this to the method for clarity (Line 612).

- Is it possible to check the correlation between performance in the YM and MWM with S1PR1 levels?;

We would also be interested in this point. The data that we have cannot reveal this for it is difficult to manipulate the S1PR1 levels by using KD and overexpression viruses.

- EM images have a poor resolution in the figures, could the authors show higher-resolution images?;

We have inserted 300 DPI images for high resolution output.

- In line 268 there is a mention of an "ShLamb1"?

We apologize for the mistake and it was revised.

**Reviewer #3 (Recommendations For The Authors):**
This study explored the role of S1P/S1PR1 signaling within the dentate gyrus (DG) in chronic pain-related memory impairment using a murine model. The authors identified decreased expression of S1PR1 in the DG of mice susceptible to memory deficits. They demonstrated that S1PR1 knockdown increased susceptibility to memory deficits, whereas its overexpression or pharmacological activation mitigated these effects. Further biochemical and immunofluorescence analyses indicated that disruptions in S1P/S1PR1 signaling were related to disruptions in actin cytoskeleton dynamics, influenced by molecular pathways involving ITGA2, Rac1/Cdc42 signaling, and the Arp2/3 complex. These findings offer intriguing insights and suggest a potential therapeutic target for treating memory impairment in chronic pain.Major Concerns:

The following five major concerns are the same with the five recommendations from Reviewer 3 on Page 9-10. Please refer to the answers above.

(1) The division of subjects into 'susceptible' and 'unsusceptible' categories requires further clarification regarding the methodologies and rationale employed, particularly concerning the use of the k-means clustering algorithm in data analysis. This explanation will strengthen the scientific grounding of the categorization process.(2) The categorization of 'susceptible' and 'unsusceptible' groups might also benefit from a more detailed analysis or discussion concerning the influence of different pain sensitivities or types of pain assessments. Although the study mentions that memory impairment stands independent of pain thresholds, a more nuanced exploration could provide deeper insights.(3) The article could benefit from more clarity on the protocol of behavioral testing, especially regarding the potential effects of repeated testing on performance outcomes due to learning or stress.(4) While the connection between S1P/S1PR1 signaling and the molecular pathways highlighted (ITGA2, Rac1/Cdc42, Arp2/3) is intriguing, only ITGA2 underwent further behavioral validation in vivo. Conducting additional behavioral assays for one or more of the molecular targets could substantially strengthen these findings.(5) Discussions regarding effective drug thresholds and the potential for non-specific effects are essential to fully evaluate the implications of pharmacological interventions utilized in the study.Minor Concerns:(1) Clarification of evidence of the specific infusion sites in pharmacological experiments would enhance the transparency and replicability of these methods.

For the infusion of S1PR1 agonist, guide cannula (internal diameter 0.34 mm, RWD) was unilaterally implanted into DG of hippocampus (-1.3 A/P, -1.95 M/L, and -2.02 D/V) as evidenced by Figure 5B.

(2) It would be beneficial if the manuscript provided details regarding the efficiency and reach of viral transfection within the neuronal population. This information would help in assessing the impact of genetic manipulations.

S1PR1 immunostaining showed that the efficiency is quite high and the reach of viral transfection is sufficient.

**Author response image 4. sa3fig4:** 

(3) The manuscript should make explicit the normalization techniques used in quantitative assessments such as Western blotting, including the housekeeping genes or proteins used for this purpose.

Here, we used housekeeping protein normalization for normalizing Western blot data. GAPDH was used as the internal control. First, the stained blot is imaged, a rectangle is drawn around the target protein in each lane, and the signal intensity inside the rectangle is measured by using ImageJ. The signal intensity obtained can then be normalized by being divided by the signal intensity of the loading internal control (GAPDH) detected on the same blot. The average of the ratios from the control group is calculated, and all individual ratios are divided by this average to obtain a new set of values, which represent the normalized values (Line 619-625).

(4) Details about the control groups in behavioral assessments were subjected to comparable handling and experimental conditions as the chronic pain groups are crucial, barring nerve injury, for maintaining the integrity of the comparative analysis.

We agree that a control group and an experimental group is identical in all respects except for one difference-nerve injury. We have added this point in the method (Line 520-522).

Minor Recommendations:

The following four minor recommendations are the same with the four minor concerns from Reviewer 3 on Page 12-13. Please refer to the answers above.

(1) Clarify the specifics of infusion site verification in pharmacological experiments.(2) Provide details on the efficiency and neuronal reach of viral transfections.(3) Explicitly describe the normalization techniques used in quantitative assessments.(4) Ensure that control groups in behavioral assessments undergo comparable handling to maintain analysis integrity.

References

(1) Gualdoni, S., et al., *Normal levels of Rac1 are important for dendritic but not axonal development in hippocampal neurons.* Biology of the Cell, 2007. **99**(8): p. 455-464.(2) Alam, M.S., *Proximity Ligation Assay (PLA).* Curr Protoc Immunol, 2018. **123**(1): p. e58.(3) Song, P., S. Zhang, and J. Li, *Co-immunoprecipitation Assays to Detect In Vivo Association of Phytochromes with Their Interacting Partners.* Methods Mol Biol, 2021. **2297**: p. 75-82.(4) Krieger, C.C., et al., *Proximity ligation assay to study TSH receptor homodimerization and crosstalk with IGF-1 receptors in human thyroid cells.* Frontiers in Endocrinology, 2022. **13**.(5) Arruda-Carvalho, M., et al., *Conditional Deletion of α-CaMKII Impairs Integration of Adult-Generated Granule Cells into Dentate Gyrus Circuits and Hippocampus-Dependent Learning.* The Journal of Neuroscience, 2014. **34**(36): p. 11919-11928.(6) Wolf, A., et al., *A Comprehensive Behavioral Test Battery to Assess Learning and Memory in 129S6/Tg2576 Mice.* PLOS One, 2016. **11**(1): p. e0147733.